# A gradient of Wnt activity positions the neurosensory domains of the inner ear

**Magdalena Żak\*, Nicolas Daudet\***

UCL Ear Institute, University College London, London, United Kingdom

**Abstract** The auditory and vestibular organs of the inner ear and the neurons that innervate them originate from Sox2-positive and Notch-active neurosensory domains specified at early stages of otic development. Sox2 is initially present throughout the otic placode and otocyst, and then it becomes progressively restricted to a ventro-medial domain. Using gain- and loss-of-function approaches in the chicken otocyst, we show that these early changes in Sox2 expression are regulated in a dose-dependent manner by Wnt/beta-catenin signalling. Both high and very low levels of Wnt activity repress Sox2 and neurosensory competence. However, intermediate levels allow the maintenance of Sox2 expression and sensory organ formation. We propose that a dorso-ventral (high-to-low) gradient and wave of Wnt activity initiated at the dorsal rim of the otic placode progressively restricts Sox2 and Notch activity to the ventral half of the otocyst, thereby positioning the neurosensory competent domains in the inner ear.

## Introduction

The inner ear is composed of several sensory organs populated with specialised mechanosensory 'hair' cells and their supporting cells. The vestibular system forms the dorsal part of the inner ear and contains the utricle, the saccule, and three semi-circular canals and their associated cristae responsible for the perception of head position and acceleration. The cochlear duct, which extends from the ventral aspect of the inner ear, contains an auditory epithelium called the organ of Corti in mammals, or the basilar papilla in birds and reptiles. All of these sensory organs originate from 'prosensory domains' that are specified in the early otocyst, a vesicle-like structure that derives from the otic placode.

The prosensory domains emerge from a population of sensory-competent cells organised in a broad antero-posterior domain along the medial wall of the otocyst. The signals controlling their specification involve a combination of cell intrinsic factors and cell-to-cell signalling pathways. The High Mobility Group (HMG) transcription factor Sox2 is considered the key prosensory factor, since its absence abolishes sensory organ formation (*Neves et al., 2007*; *Kiernan et al., 2005*; *Pan et al., 2013*). Sox2 is initially expressed in a large portion of the otocyst and then becomes progressively restricted to two distinct prosensory domains in its anterior and posterior regions (*Steevens et al., 2017*). The anterior domain is neuro-sensory competent: it forms several vestibular sensory epithelia (the anterior and lateral crista; the macula of the utricle) by segregation and the otic neuroblasts, which delaminate from the epithelium to differentiate into the auditory and vestibular neurons (*Steevens et al., 2017*; *Mann et al., 2017*; *Morsli et al., 1998*; *Adam et al., 1998*; *Fritzsch et al., 2002*; *Satoh and Fekete, 2005*). The posterior one, in contrast, is non-neurogenic and thought to generate the posterior crista only. The auditory sensory patch (the organ of Corti in mammal or basilar papilla in birds) is specified after the vestibular organs (*Morsli et al., 1998*) but the otic territory from which it derives is ill-defined.

The factors regulating Sox2 expression during early otic development are likely to play a key role in the control of the timing and spatial pattern of sensory organ formation. For example, Notch-mediated lateral induction, dependent on the Notch ligand Jagged 1 (Jag1), is required for the

**\*For correspondence:**
m.zak@ucl.ac.uk (MZ);
n.daudet@ucl.ac.uk (ND)

**Competing interests:** The authors declare that no competing interests exist.

maintenance of Sox2 expression and sensory organ formation (*Kiernan et al., 2001*; *Kiernan et al., 2006*; *Brooker et al., 2006*; *Daudet et al., 2007*; *Pan et al., 2010*; *Tsai et al., 2001*). Forcing Notch activity leads to the formation of ectopic sensory patches (*Pan et al., 2013*; *Pan et al., 2010*; *Daudet and Lewis, 2005*; *Hartman et al., 2010*; *Neves et al., 2011*), suggesting that it is one of the key factors maintaining Sox2 and prosensory competence (reviewed in *Daudet and Żak, 2020*). Another candidate regulator of prosensory specification is Wnt signalling (reviewed in *Żak et al., 2015*), which relies on interactions between soluble Wnt ligands and their transmembrane Frizzled receptors to activate canonical and non-canonical branches of the Wnt pathway (*Komiya and Habas, 2008*). Beta-catenin (β-catenin) is the key element in the intracellular cascade of canonical Wnt signalling. When Wnt signalling is active, β-catenin escapes degradation and moves to the nucleus where it interacts with transcription factors to regulate the expression of Wnt target genes. Previous studies have shown that Wnt activity is elevated in the dorsal aspect of the otocyst and is required for vestibular system morphogenesis (*Riccomagno et al., 2005*; *Noda et al., 2012*), but its role in the context of prosensory specification is still unclear.

In this study, we investigated the roles of Wnt signalling in the embryonic chicken otocyst and its potential interactions with Notch signalling. We show that a (high to low) gradient of Wnt activity exists along the dorso-ventral axis of the otocyst. By co-transfecting reporters of Notch or Wnt activity with modulators of these pathways, we found that manipulation of Notch activity does not impact on Wnt signalling. In contrast, high levels of Wnt activity repress neurosensory specification and Jag1-Notch signalling. The consequences of reducing Wnt signalling were strikingly different along the dorso-ventral axis of the otocyst: in dorsal regions, it induced ectopic neurosensory territories, whilst in the ventral domains, it repressed Sox2 expression, suggesting that low levels of Wnt activity are required for prosensory specification. Using pharmacological treatments in organotypic culture of otocysts, we confirmed that decreasing Wnt activity triggers expansion of prosensory genes Jag1 and Sox2 dorsally and reduces their expression in the central part of the ventral territories. Furthermore, in ovo reduction of Wnt activity can also trigger delamination of ectopic neuroblasts from the otocyst. Altogether, these data suggest that a dorso-ventral gradient of Wnt signalling acts upstream of Notch to position, in a dose-dependent manner, the neurosensory-competent domains of the otocyst.

## Results

### Canonical Wnt activity forms a dorso-ventral gradient in the otocyst and is reduced in neurogenic and prosensory domains

To examine the spatial pattern of Wnt activity during early prosensory specification, we electroporated the otic cup of E2 chicken embryos with a Wnt reporter plasmid 5TCF::H2B-RFP, containing 5 TCF binding sides (upstream of a minimal TK promoter) regulating the expression of a red fluorescent protein fused with Histone 2B (H2B-RFP) (*Figure 1a*). A control EGFP expression vector was co-electroporated to visualise all transfected cells. In all of the samples analysed at E3 (n > 12), RFP expression was confined to the dorsal 2/3 of the otocyst (*Figure 1b–b′*) on both medial and lateral walls (*Video 1*). Wnt ligands can diffuse and elicit spatial gradients of Wnt activity in some tissues (*Pani and Goldstein, 2018*; *Farin et al., 2016*). To test if this might be the case in the otocyst, we quantified reporter fluorescence intensity in individual cell nuclei according to their X and Y coordinates (*Figure 1c*, total of 7322 cell in seven samples). The results showed that cells with high Wnt activity occupy the dorso-posterior domain of the otocyst and those with low (or no) activity its ventral portion (*Figure 1c*). To confirm the presence of this Wnt gradient, we calculated the median intensity values of groups of nuclei located at 10 different levels along the dorso-ventral axis (*Figure 1d*). The plot revealed a relatively linear decrease in fluorescence, suggesting that cells located at different dorso-ventral positions are exposed to distinct Wnt activity levels.

At E2–E3, Notch is active in the anterior neurosensory competent domain of the otocyst where it regulates the production of otic neuroblasts by lateral inhibition. To examine the relation between the spatial patterns of Wnt and Notch activities, we co-electroporated fluorescent Wnt and Notch reporters together with a control plasmid driving expression of 3xnls-mTurquoise2 (a blue fluorescent protein) in the E2 otic cup (*Figure 1e*). The Notch reporter T2-Hes5::nd2EGFP consisted of a mouse Hes5 promoter driving the expression of a nuclear and destabilised EGFP

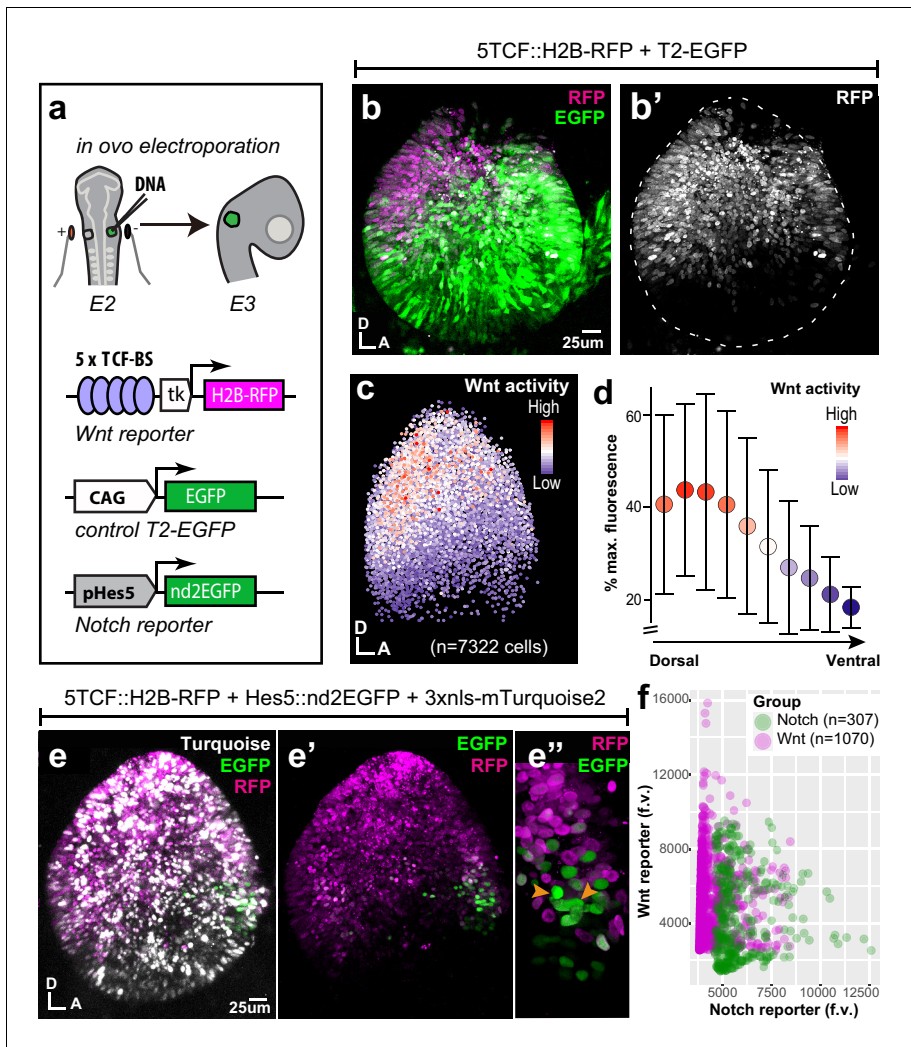

**Figure 1.** Spatial pattern of Wnt activity in the E3 chicken otocyst. In all panels, dorsal (D) is up and anterior (A) is right. (**a**) E2 chicken embryos were co-electroporated either with Wnt reporter and a control plasmid T2-EGFP or Wnt reporter together with a Notch reporter and collected at E3. The Wnt reporter (5TCF::H2B-RFP) contains 5 TCF/LEF binding sites regulating an H2B-RFP fusion protein. In the Notch reporter (T2-Hes5::nd2EGFP), the mouse Hes5 promoter regulates expression of a nuclear destabilised EGFP. The control vector drives constitutive expression of EGFP. (**b–b'**) Whole-mount view of an E3 otocyst electroporated with the Wnt reporter and a control plasmid. Wnt-responsive cells (**b'**) are detected in the dorsal 2/3 of the otocyst. (**c**) Quantification of Wnt reporter fluorescent levels in individual cells from seven otocysts transfected with the Wnt reporter (see Materials and methods). A decreasing gradient of Wnt reporter fluorescence is observed along the dorso-ventral and postero-anterior axis of the otocyst. (**d**) Plot of the normalised median fluorescence levels of cells as a function of their position along the dorso-ventral axis of the otocyst. The standard deviation bars reflect variability in fluorescent intensity along the anterio-posterior axis. (**e–e"**) E3 chicken otocyst co-electroporated with the Wnt and Notch reporters and a control plasmid. The Notch reporter marks the prosensory cells in the antero-ventral prosensory domain (**e"**). (**f**) A representative scatter plot of the mean fluorescence values (f.v.) for Wnt (5TCF::H2B-RFP) and Notch (T2-Hes5::nd2EGFP) reporters in individual cells of the anterior prosensory domain. The two groups correspond to cells segmented using either the Notch (green) or Wnt (magenta) reporter fluorescence signal. The cells with high Notch activity tend to have low levels of Wnt activity, and cells with high Wnt activity have low levels of Notch activity, but there is no inverse correlation between the reporters activities at intermediate fluorescence intensity values.

The online version of this article includes the following source data for figure 1:

**Source data 1.** Wnt reporter activity in the E3 chicken otocyst.

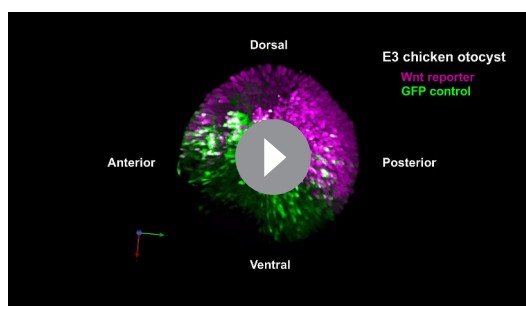

**Video 1.** E3 chicken otocyst electroporated with Wnt reporter 5TCF::H2B-RFP and control plasmid T2-EGFP. https://elifesciences.org/articles/59540#video1

(*Chrysostomou et al., 2012*). In samples collected 24 hr after electroporation, we observed an overlap between Wnt and Notch reporters at the dorsal border of the anterior neurosensory-competent domain (n > 5) (*Figure 1e′–e″*). However, the quantification of the mean fluorescence levels of both reporters in single cells revealed an inverse correlation between the fluorescence levels of Wnt and Notch reporters within transfected cells (*Figure 1f*) suggesting a potential antagonism between Wnt and Notch activity.

## Wnt activity antagonises notch signalling in the otocyst

To test the interactions between Wnt and Notch signalling, we used gain- (GOF) and loss-of-function (LOF) β-catenin constructs: a full-length constitutively active β-catenin carrying the S35Y mutation (βcat-GOF) to induce Wnt activity; a truncated form of β-catenin composed of the Armadillo domain only to block Wnt (βcat-LOF) (*Figure 2a*). To validate their effects, we co-transfected these with the Wnt reporter at E2 and examined the otocysts 24 hr later. Compared to control conditions (*Figure 2b–b′*), βcat-GOF led to a clear expansion of the Wnt reporter fluorescence in the ventral otocyst (n = 6) (*Figure 2c–c′*). Conversely, overexpressing βcat-LOF restricted Wnt reporter fluorescence (n = 4) to the most dorsal territories of the otocyst (*Figure 2d–d′*), suggesting a strong reduction in Wnt activity levels. Having confirmed the ability of these constructs to activate or reduce canonical Wnt signalling, we next examined their impact on Notch activity.

In control experiments, the Notch reporter T2-Hes5::nd2EGFP was activated in the anterior neurosensory domain, with few cells with weaker Notch activity present in the posterior prosensory domain (*Figure 2e–e′*). After co-electroporation with the βcat-LOF construct, Notch activity expanded beyond the prosensory domains and in the dorsal otocyst (n = 5) (*Figure 2f–f′*). In contrast, co-electroporation with the βcat-GOF construct strongly decreased Notch activity so that only a few Notch-active cells were detected within the anterior domain (n = 5) (*Figure 2g–g′*). To test if Notch activity could reciprocally regulate Wnt signalling, we co-transfected the Wnt reporter with constructs previously shown to activate (chicken Notch one intracellular domain or NICD1, see *Daudet and Lewis, 2005*) or block (dominant-negative form of human Mastermind-like1 or DN-MAML1, see *Maillard et al., 2004*) Notch activity. The intensity or spatial pattern of activation of the Wnt reporter in response to manipulations in Notch activity remained very similar to that of controls (*Figure 2—figure supplement 1*). Altogether, these results show that canonical Wnt signalling antagonises Notch activity in the otocyst, whilst Notch does not appear to affect the levels and spatial pattern of Wnt activity.

## Genetic manipulation of Wnt activity disrupts prosensory specification in a location-specific manner

We next tested the effects of manipulating Wnt activity on prosensory specification. Samples electroporated at E2 were collected at E4, then immunostained for Jag1 and Sox2. In controls, Jag1 and Sox2 were detected in the anterior and posterior prosensory domains and within a U-shaped band of cells extending in between these two domains in the ventral half of the otocyst (*Figure 3a–a″*). The overexpression of βcat-GOF reduced, in a cell-autonomous manner, the levels of Jag1 and Sox2 expression in the majority of transfected prosensory cells but did not induce any change in the dorsal region of the otocyst (n = 6) (*Figure 3b–c″*, *Figure 3—figure supplement 1a–d*). In contrast, βcat-LOF induced the formation of ectopic prosensory patches in the dorsal otocyst (n = 6) (*Figure 3d–d″*). All ectopic patches were positive for Sox2, but only some expressed Jag1 (*Figure 3e–e″*). The ability of βcat-LOF to induce ectopic prosensory territories dorsally was dependent on functional Notch signalling. In fact, very few ectopic patches formed in samples co-transfected with βcat-LOF and DN-MAML1 (*Figure 3—figure supplement 2a–b′″*). In the ventral otocyst, however, the consequences of decreasing Wnt signalling were radically different: βcat-LOF

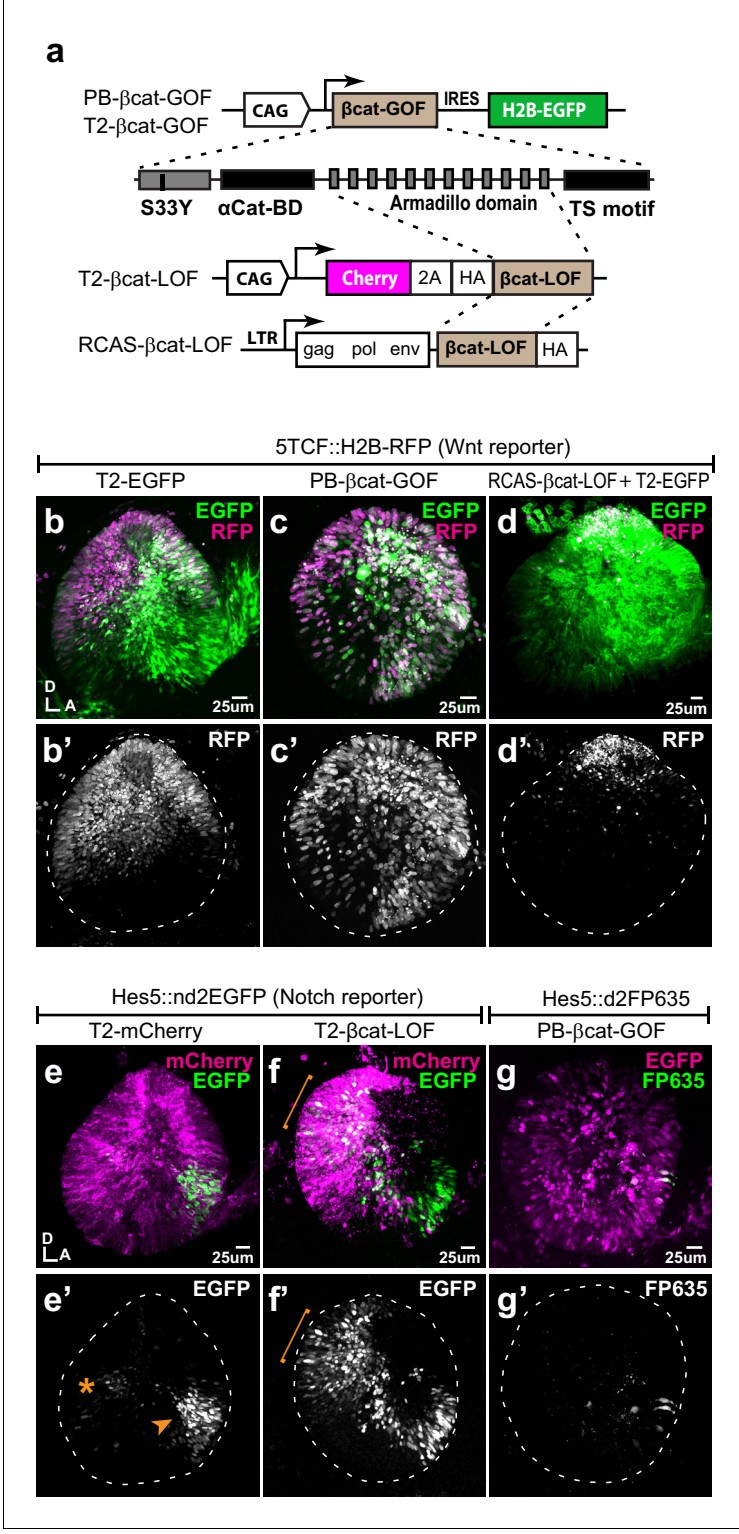

**Figure 2.** Wnt signalling antagonises Notch activity. (a–d') Schematic representation of the Piggybac, Tol2, and RCAS constructs used for β-catenin gain- (GOF) and loss-of-function (LOF) experiments. The PB-βcat-GOF and T2-βcat-GOF contain the full-length β-catenin including the α-catenin binding domain (αCat-BD), 12 Armadillo domains, the transactivator (TS) motif, and the S33Y mutation preventing phosphorylation and degradation. The RCAS-βcat-LOF and T2-βcat-LOF constructs drive expression of a truncated form of β-catenin comprising the Armadillo repeats only. (b–d') Activity of the Wnt reporter in E3 otocysts co-electroporated with either T2-EGFP (control; b–b'), PB-βcat-GOF (c–c'), or RCAS-βcat-LOF (d–d'). Note the ventral expansion of the Wnt reporter

*Figure 2 continued on next page*

*Figure 2 continued*

fluorescence in (c–c') and its restriction to the most dorsal part of the otocyst in (d–d'). (e–g') Activity of the Notch reporters T2-Hes5::nd2EGFP or Hes5::d2FP635 in E3 otocysts co-electroporated with either T2-mCherry (control, e–e'), T2-βcat-LOF (f–f'), or PB-βcat-GOF (g–g'). The Notch reporter is normally activated in the anterior (arrowhead) and to a lesser extent posterior (asterisk) prosensory domains of the otocyst (e–e'). It is strongly upregulated in dorsal regions transfected with the T2-βcat-LOF construct (brackets in f–f'), but barely detectable in otocysts co-electroporated with PB-βcat-GOF (g–g'). On the other hand, manipulation of Notch activity had no discernible effect on the activity of the Wnt reporter (*Figure 2—figure supplement 1*).

The online version of this article includes the following figure supplement(s) for figure 2:

**Figure supplement 1.** Manipulating Notch activity does not affect Wnt signalling.

transfected cells exhibited a loss of Sox2 expression (*Figure 3f–g''*), suggesting a loss of prosensory character (n = 6). To investigate the potential effects of Wnt signalling on otic neurogenesis, otic cups were electroporated with either control or βcat-LOF constructs, collected at E4, then immunostained for Islet1, a LIM homeobox transcription factor expressed by otic neuroblasts. In all otocysts,

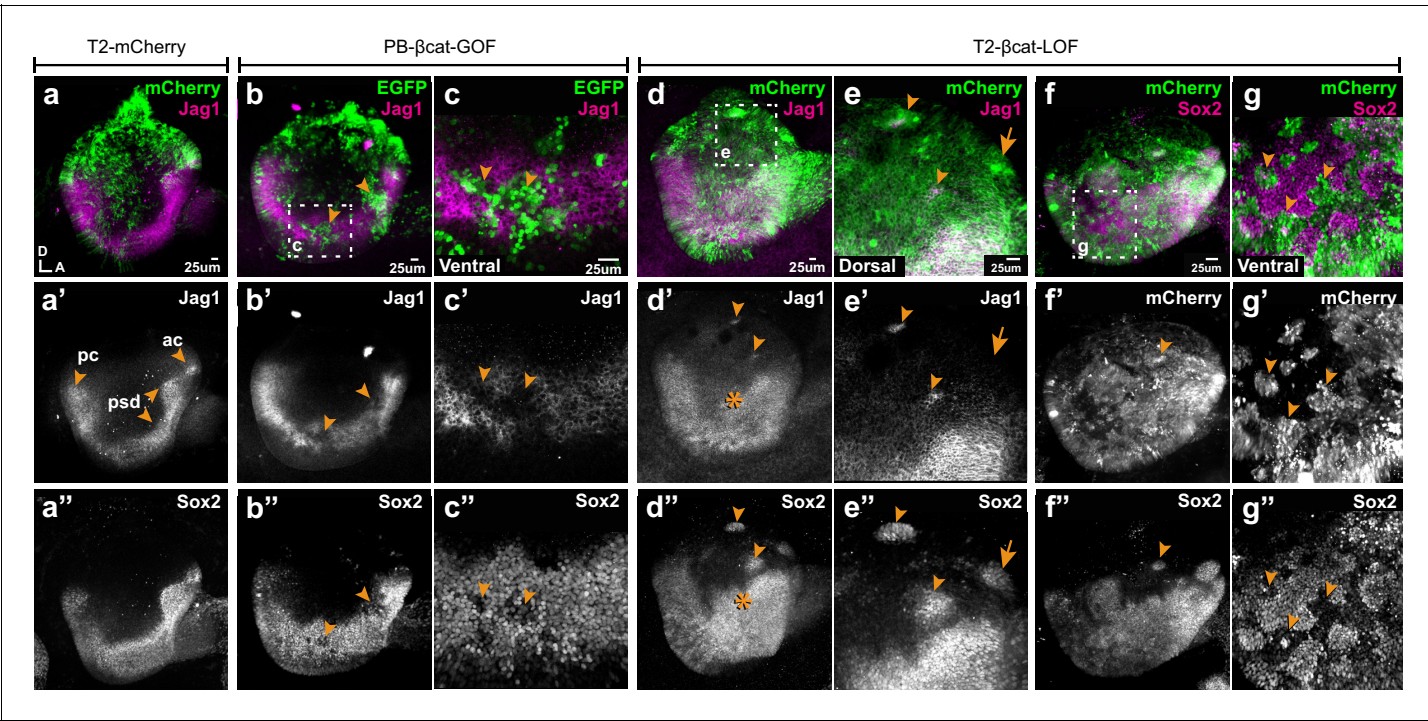

**Figure 3.** Wnt signalling antagonises prosensory specification. Whole-mount views of E4 chicken otocysts electroporated at E2 and immunostained for Jag1 and Sox2 expression. (a–a'') Control sample electroporated with T2-mCherry. Jag1 and Sox2 are expressed in a U-shaped ventral common prosensory-competent domain (psd) and prospective prosensory domains (pc = posterior crista; ac=anterior crista). (b–c'') βcat-GOF overexpression induces a mosaic down-regulation of Sox2 and Jag1 expression (arrowheads) in the ventral half of the otocyst. High magnification views of transfected cells (arrowheads in c–c'') and analysis of mean fluorescence values of Sox2 and βcat-GOF (*Figure 3—figure supplement 1a–e*) show that this effect is cell-autonomous. (d–g'') Otocysts transfected with T2-βcat-LOF exhibit a dorsal expansion of Jag1 and Sox2 expression (star in d'–d'') and ectopic prosensory patches dorsally (arrowheads in d'–d'', f'') and high magnification views in (e–e''). Note that some ectopic Sox2-positive patches are Jag1-negative (arrows in e'–e''). The prosensory effects of βcat-LOF were dependent on Notch activity (*Figure 3—figure supplement 2a–b'''*). In contrast, in the ventral-most aspect of the otocyst, βcat-LOF overexpressing cells exhibit reduced Sox2 expression (arrowheads in high magnification views g–g''). Overexpression of βcat-LOF elicits the formation of ectopic Islet-1 expressing otic neurons in the posterior and dorsal aspect of the otocyst (*Figure 3—figure supplement 3*).

The online version of this article includes the following figure supplement(s) for figure 3:

**Figure supplement 1.** Analyses of Sox2 (magenta) and βcat-GOF (EGFP, green) fluorescence intensity levels in transfected prosensory regions.

**Figure supplement 2.** Effects of simultaneous loss of Wnt and Notch activity on prosensory specification.

**Figure supplement 3.** Blocking Wnt signalling triggers ectopic neurogenesis.

Islet1 was strongly expressed in the neuroblasts delaminating from the anterior neurogenic patch (*Figure 3—figure supplement 3a–b'*). However, in otocysts electroporated with the βcat-LOF construct, we noticed that some transfected cells were clustered outside of the epithelial lining of the dorsal and posterior otocyst (n = 3/4) (*Figure 3—figure supplement 3a–a'*). These cells expressed Islet1 (*Figure 3—figure supplement 3b–c'*), which strongly suggests that they are ectopic delaminating otic neuroblasts. This result indicates that Wnt signalling regulates both prosensory and neuronal specification in the otocyst.

To assess the long-term consequences of these manipulations on sensory organ formation, we incubated some of the embryos electroporated at E2 with transposon vectors (allowing stable integration of the transgenes) until E7, a stage when individualised sensory organs can be easily identified. Transfected inner ears were immunostained for Sox2 and two proteins expressed in differentiated hair cells: Myosin7a (Myo7a), an unconventional myosin expressed in hair cell cytoplasm and the hair cell antigen (HCA), a protein tyrosine phosphatase receptor expressed in hair cell bundles (*Gibson et al., 1995*; *Goodyear et al., 2003*). In control EGFP-transfected samples, inner ear morphology was normal and hair cells were detected in the vestibular organs (the saccule, utricle, and the three cristae) but not in the basilar papilla extending within the ventral cochlear duct (*Figure 4a–a''*). Severe malformations were observed upon transfection with the βcat-GOF construct: four out of five samples analysed lacked some of the vestibular organs; the remaining patches were small and abnormally shaped but populated with hair cells (*Figure 4b–c''*). The basilar papilla was either shortened or missing in four samples. Surprisingly, the EGFP signal of the two βcat-GOF constructs (cloned in Piggybac and Tol2 vectors) tested for these experiments was seen 24 hr post-electroporation but not at E7, suggesting that the transfected cells might have been eliminated from the epithelium by this stage (*Figure 4b,c*). Long-term overexpression of βcat-LOF (using RCAS or Tol2 vectors) severely altered the morphogenesis of the inner ear (n > 6) (*Figure 4d–d''*). Many ectopic Sox2-positive patches of various sizes occupied the dorsal region of the inner ear (*Figure 4d–d'*). These were populated by hair cells (*Figure 4e–e''*), suggesting that the ectopic (dorsal) prosensory patches observed at E4 in βcat-LOF conditions can differentiate into mature sensory territories. In contrast, the loss of Wnt activity in ventral regions blocked the formation of the cochlear duct and basilar papilla (*Figure 4d–d''*). The only Sox2-positive cells remaining in the ventral region of the inner ear were untransfected; they formed small patches surrounded by Sox2-negative cells transfected with the βcat-LOF construct (*Figure 4f–f''*).

Altogether, these results show that the early and sustained manipulation of Wnt activity affects the formation of the sensory organs and inner ear morphogenesis. The overactivation of Wnt signalling antagonises prosensory specification and may compromise long-term cell viability. On the other hand, reducing Wnt activity induces prosensory character dorsally, whilst in ventral regions it represses it. In light of the endogenous high-to-low gradient of canonical Wnt activity along the dorso-ventral axis of the otocyst, one possible explanation for these dual effects is that Wnt signalling regulates prosensory specification in a dose-dependent manner: high levels of canonical Wnt activity (dorsally) repress it, but low levels (ventrally) are however necessary for cells to acquire or maintain their prosensory character.

## Wnt activity is maximal in dorsal and non-sensory territories of the developing inner ear

To gain further insights into the temporal and spatial relationship between canonical Wnt activity and prosensory specification, we electroporated the otic placode of E2 (stage HH10) embryos with the Wnt reporter 5TCF::H2B-RFP and collected the samples at 6 hr (stage HH11), 12 hr (HH12), 24 hr (HH18), and 3 days (E5, HH26-27) post-electroporation. At HH11, Sox2 staining was detected throughout the otic placode but decreased in intensity towards its dorso-anterior side; a few cells with low Sox2 expression were also positive for the Wnt reporter in the dorsal rim of the otic placode (arrowheads in *Figure 5a–a'''*). At HH12, Sox2 staining was confined to the ventral half of the otic cup, with the strongest expression in the anterior prosensory domain (star in *Figure 5b''*). The Wnt reporter was detected in the dorsal side in a complementary manner to Sox2 expression (arrows in b') and it overlapped with Sox2 at the dorsal limit of the prosensory domain (arrowheads in *Figure 5b–b'''*). As previously described, Wnt reporter activity was detected in the dorsal half of the HH18 (E3) otocyst, whilst Sox2 was confined to its ventral half; only a few cells at the dorsal edge of the prosensory domain were positive for the Wnt reporter and Sox2 (arrowheads in *Figure 5c'–c'''*).

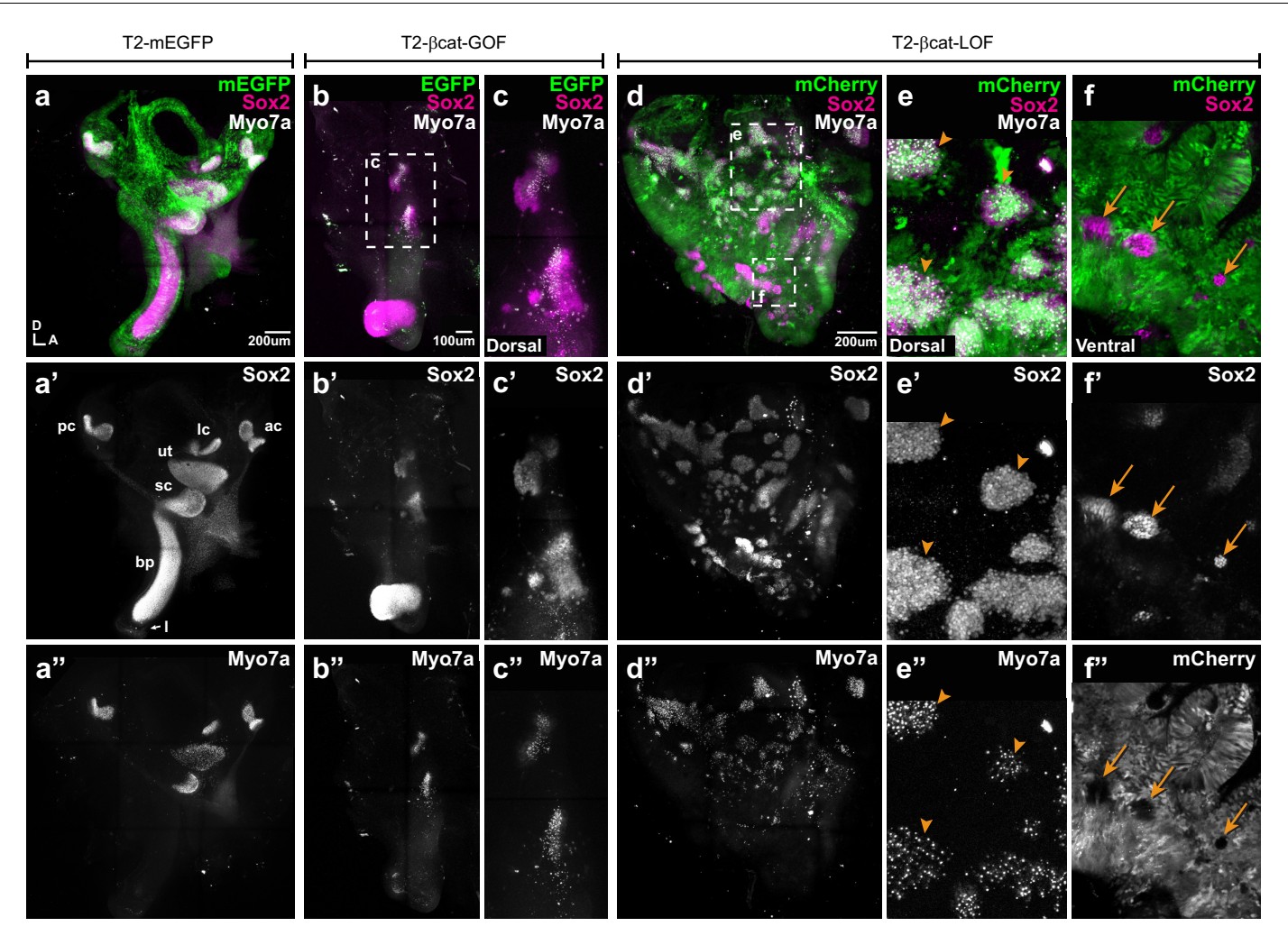

**Figure 4.** Manipulating Wnt activity alters inner ear sensory organ formation. (a–a") Whole-mount (tiled maximum projection) views of an E7 chicken inner ear electroporated at E2 with a control vector (T2-mEGFP) and immunostained for Sox2 (a') and two hair cell markers, Myo7a and HCA ('HC' in all panels). (a") All sensory organs are properly formed: posterior (pc), anterior (ac) and lateral (lc) cristae, saccule (sc), utricle (ut), basilar papilla (bp), and lagena (l). (b–c") An inner ear transfected with T2-βcat-GOF. Note the absence of EGFP expression and severe defects in overall morphology of the vestibular system and basilar papilla; the remaining sensory patches are small and abnormally shaped (b'). (c–c") Higher magnification of the vestibular Sox2-positive patches containing Myo7a and HCA-expressing hair cells. (d–f") An inner ear transfected with T2-βcat-LOF. (d–d") Whole-mount (tiled maximum projection) views demonstrating the presence of numerous ectopic sensory patches with hair cells, and severe defects in inner ear morphology. (e-e') Higher magnification of the dorsal region, where transfected cells form ectopic sensory patches positive for Sox2 (e') and populated with Myo7a and HCA-expressing hair cells (arrowheads). (f–f") In contrast, in ventral domains, EGFP-positive patches are devoid of Sox2 and hair cell markers expression. The only remaining Sox2-expressing patches are not transfected (arrows).

In order to study the pattern of Wnt activity at later stages of inner ear development, we generated a Tol2-Wnt reporter (T2-5TCF::nd2Scarlet) containing the same regulatory elements and controlling the expression of a nuclear and destabilised form of the Scarlet red fluorescent protein (*Bindels et al., 2017*). At E5, the Tol2-Wnt reporter fluorescence remained elevated in the dorsal aspect of the inner ear containing the vestibular organs, the semi-circular canals and the endolymphatic duct (*Figure 5d-d''*). Reporter activity was highest in the non-sensory tissues surrounding the sensory organs, which at this stage have partially segregated from one another (*Figure 5e–e'''*). There were however a few Sox2-positive cells with comparatively low levels of reporter activity within the cristae and utricle (arrow in *Figure 5e'*). In the ventral aspect of the inner ear, the cochlear duct was largely devoid of Wnt reporter activity with the exception of the distal tip of the basilar papilla,

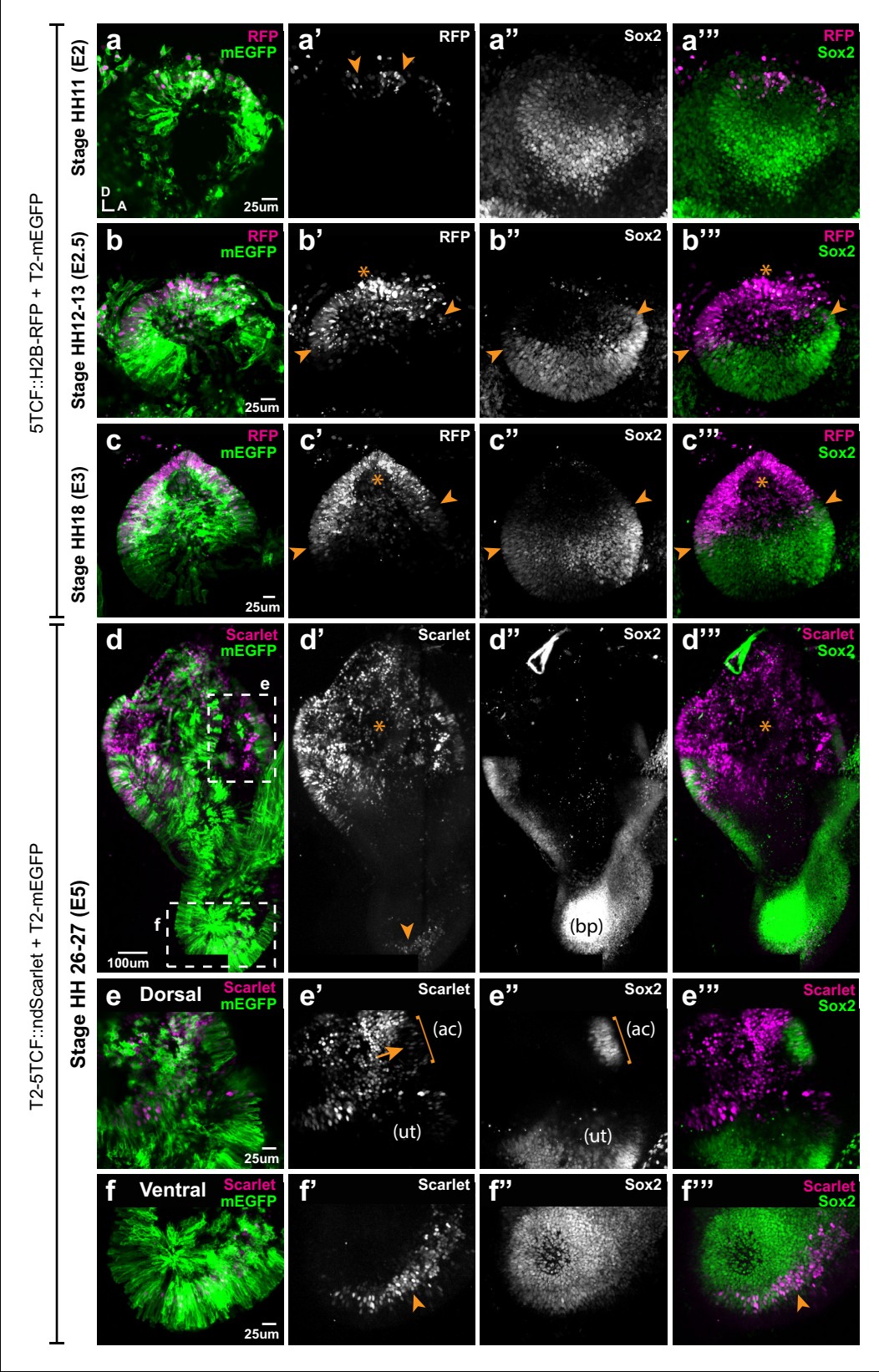

**Figure 5.** Spatial pattern of Wnt activity in the developing chicken inner ear. Samples co-electroporated at the early otic placode stage with a Wnt reporter (5TCF::H2B-RFP or T2-5TCF::nd2Scarlet for long-term integration) and a control plasmid (T2-mEGFP in **a–d**) were collected 6 hr (stage HH11), 12 hr (HH12-13), 24 hr (HH18), and 3 days

*Figure 5 continued on next page*

*Figure 5 continued*

(E5, HH26-27) post-electroporation and immunostained for Sox2 expression. (a–a′″) Only a few cells on the dorso-medial wall of the otic placode are positive for Wnt reporter 5TCF::H2B-RFP (arrowheads in a′), whilst most otic cells express Sox2 (a″). (b–c′″) Wnt activity increases gradually in the dorsal aspect of the otic cup and otocyst (stars in b′ and c′), concomitant to a dorsal decrease in Sox2 expression (b″–c″). Note the overlap between the signals of Wnt reporter and Sox2 (arrowheads in b′–b″ and c′–c″) at the dorsal edges of the prosensory domains. (d–d′″) The Wnt reporter T2-5TCF::nd2Scarlet is strongly active in the dorsal half of the E5 inner ear (stars in d′–d′″) and a weaker signal is also detected at the tip of developing basilar papilla (arrowhead). (e–e′″) Higher magnification views of the anterior vestibular organs. Note the high levels of Wnt activity in the non-sensory territories. In comparison, transfected prosensory cells located within the anterior crista (ac) and utricle (ut) have lower levels of fluorescence (arrow in e′). (f–f′″) Higher magnification views of the ventral (distal) tip of the basilar papilla, which also contains Wnt-active prosensory cells (arrowheads).

which consistently contained Sox2-expressing cells with relatively low levels of Wnt reporter fluorescence (*Figure 5f–f′″*).

In summary, these results show that the gradient of Wnt activity observed at the otocyst stage is established progressively in a dorso-ventral manner from the otic placode stage and maintained at later stages of inner ear development. Remarkably, the dorsal suppression of Sox2 expression in the otic cup coincides with the upregulation of Wnt activity, which fits with the idea that high levels of Wnt signalling antagonise prosensory character.

## Wnt activity regulates the positioning of neurosensory-competent domains

We next explored the effects of known modulators of Wnt activity on the spatial pattern of Sox2 and Jag1 expression. We first cultured E3 chicken otocysts in control medium or medium supplemented with lithium chloride (LiCl), which promotes canonical Wnt signalling by repressing GSK3β activity (*Klein and Melton, 1996*). In samples that had been previously electroporated at E2 with the Wnt reporter, a 24 hr treatment with LiCl induced an upregulation of the activity of the reporter in ventral territories of the otocyst (*Figure 6a–b′*). However, LiCl treatments (5–35 µM) did not abolish Sox2 expression but caused a dose-dependent shift of the position and orientation of the Sox2-positive prosensory domain towards the antero-ventral aspects of the otocyst (n = 5–7 per concentration) (*Figure 6—figure supplement 1a–e*). This suggests that increasing endogenous Wnt activity might repress prosensory specification in the ventro-posterior otocyst, but that the requirements for low levels of Wnt activity for prosensory specification may be limited to a developmental window before E3–4. We next tested the consequences of decreasing Wnt signalling by treating E3 chicken otocysts with IWR-1, a tankyrase inhibitor that stabilises Axin2 (*Chen et al., 2009*), a member of the β-catenin degradation complex. Compared to controls, IWR-1 treatment (300 µM) induced a strong reduction of the Wnt reporter in E3 otocysts cultured for 24 hr (*Figure 6c–c′*). We next compared the expression of Sox2 and Jagged1 in E3 otocysts (n = 4) maintained for 24 hr in either IWR-1 or DMSO. In controls, strong staining for both markers was detected in the anterior domain and weaker expression was present towards the posterior prosensory domain (*Figure 6d–d″*). In contrast, in samples treated with IWR1, Sox2 and Jag1 expression was markedly expanded in the dorsal half of the otocysts (*Figure 6e–e″*) and somewhat reduced in a vertical ventral domain located in the middle of the otocyst. Altogether, these results confirmed that Wnt activity represses prosensory specification and supported the hypothesis that the spatial pattern and levels of Wnt activity regulate the positioning of the prosensory territories of the inner ear.

## Discussion

The axial patterning of the otocyst is regulated by the interactions between cell-intrinsic 'fate determinants' and the signalling pathways directing their expression to specific otic territories (*Fekete and Wu, 2002*). In this context, Wnt signalling has been proposed to act as an essential dorsalising factor. In fact, the dorsal hindbrain produces Wnt1 and Wnt3a, which are thought to trigger high Wnt activity and the expression of vestibular-specific genes in the dorsal otocyst (*Riccomagno et al., 2005*; *Noda et al., 2012*; *Bok et al., 2005*). In compound *Wnt1/Wnt3a* null

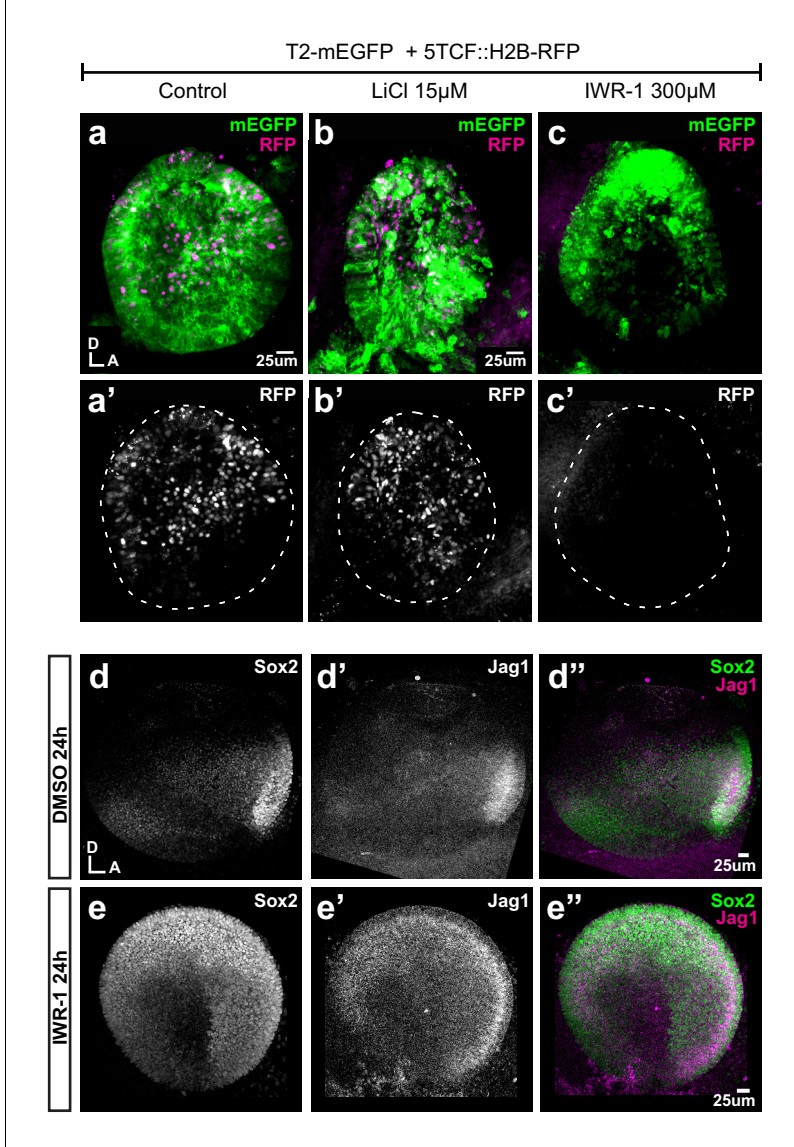

**Figure 6.** Pharmacological modulation of Wnt activity in explanted E3 otocysts. (**a–c'**) Whole-mount views of otic cups co-electroporated with the Wnt reporter and a control EGFP vector and incubated for 24 hr in control medium (DMSO) (**a–a'**), or media supplemented with either the Wnt agonist LiCl (**b–b'**) or the antagonist IWR-1 (**c–c'**). (**d–e''**) E3 otocysts cultured for 24 hr in IWR-1 or DMSO (vehicle) as a control. IWR-1 treatment results in a dorsal expansion of Sox2 and Jag1 staining. The effects of increasing concentration of LiCl on Sox2 expression are shown in *Figure 6—figure supplement 1a–e*.

The online version of this article includes the following figure supplement(s) for figure 6:

**Figure supplement 1.** Effects of the Wnt agonist LiCl on Sox2 expression.

---

mice or *β-catenin* null mice, the entire vestibular system fails to form and a poorly developed cochlear-like canal is the only remaining inner ear structure (*Riccomagno et al., 2005*). However, the specific roles of Wnt signalling in the formation of inner ear sensory organs remain unclear. In this study, we took advantage of the amenability of the chicken embryo to mosaic manipulation of gene expression to uncover new roles for Wnt signalling in prosensory and neuronal specification in the inner ear.

## A dorso-ventral wave and gradient of canonical Wnt activity regulates the spatial pattern of otic neurosensory competence

Previous studies in transgenic mice harbouring TCF/Lef reporters *Riccomagno et al., 2005*; *Noda et al., 2012* have shown that canonical Wnt is active in the dorsal otocyst. Our results with a fluorescent TCF/Lef reporter confirm these findings but also show a dorsal-to-ventral (and to some extent posterior-to-anterior) linear reduction in the fluorescence levels of individual cells in the chicken otocyst. This gradient could reflect differences in both dosage of, and total exposure time to, Wnt activity. In fact, dorsal cells are the closest to the hindbrain, which is the proposed source of Wnt ligands influencing otic patterning, but they are also the first to experience Wnt activity during inner ear development. Our data show that this Wnt gradient regulates the expression of Sox2, an essential factor for prosensory specification.

Recent studies (*Steevens et al., 2017*; *Steevens et al., 2019*; *Gu et al., 2016*) have uncovered dynamic changes in Sox2 expression pattern in the early otic vesicle, which were also apparent in our experiments: as the otic placode transforms into a vesicle, Sox2 is confined to the ventral half of the otocyst. Strikingly, the ventral expansion of the Wnt-active domain coincided in space and time with the dorsal down-regulation of Sox2. Our GOF and LOF studies strongly suggest that canonical Wnt drives this ventral restriction in a cell-autonomous manner. In fact, overexpressing βcat-GOF inhibits Sox2 expression and prosensory patch formation ventrally. Conversely, the blockade of canonical Wnt resulting from the overexpression of βcat-LOF at E2 leads to the formation of ectopic Sox2/Jag1-positive prosensory patches in the dorsal otocyst. At least some of these are neurogenic, as confirmed by the presence of delaminating Islet1-positive neuroblasts. A comparable result was obtained in E3 otocysts treated in vitro with the Axin2 stabiliser IWR-1, which exhibited a dorsal upregulation of Sox2/Jag1 expression. In contrast, in the ventral otocyst, βcat-LOF transfected cells had much reduced levels of Jag1 and Sox2, suggesting a loss of prosensory character. In IWR-1 treated otocyst, the spatial pattern of Jag1 and Sox2 expression was only partly reduced within the ventral domain, possibly due to the fact that some of the ventral cells might already be irreversibly committed to a prosensory fate at E3. Altogether, these results imply that high levels of Wnt activity repress Sox2 and neurosensory specification, but transient or low levels of Wnt activity are required for this process to occur. We propose that these dose-dependent effects, elicited by the dorso-ventral wave and gradient of Wnt activity, confine neurosensory-competent domains to the ventral aspect of the otocyst (*Figure 7*).

Previous studies investigating the roles of Wnt signalling in the early developing inner ear have focused on its requirement for the morphogenesis of the non-sensory structures of the vestibular system. However, one study using tamoxifen-inducible deletion of β-catenin in the mouse embryo reported some defects supporting our conclusions: supressing β-catenin expression at E10.5 led to a reduction in hair cell formation within some vestibular organs at E14.5, consistent with a requirement for prosensory specification (*Rakowiecki and Epstein, 2013*). A major difference with our results is that ectopic sensory patches did not form dorsally in the mouse otocyst. This is most likely explained by the fact the *β-catenin* cKO allele is a complete null, whilst the overexpression of truncated β-catenin could lead to a partial LOF. In the dorsal otocyst, where endogenous Wnt activity is the strongest, βcat-LOF could reduce Wnt activity to a level that becomes permissive for the maintenance of Sox2 expression.

Strikingly, *Rakowiecki and Epstein, 2013* also found that overexpressing an active form of β-catenin (lacking exon 3) in the embryonic mouse inner ear induces a loss of prosensory markers and hair cells in the anterior and posterior cristae. This result was at the time surprising, since an earlier study by *Stevens et al., 2003* suggested that forcing Wnt activation in the chicken inner ear elicits the formation of ectopic sensory territories. The N- and C-terminal truncated form of β-catenin (containing the Armadillo repeats only, or βcat-LOF in our experiments) used by *Stevens et al., 2003* was thought to be a GOF protein, since it can induce axis duplication as efficiently as the full-length β-catenin protein in *Xenopus* embryos (*Funayama et al., 1995*). However, we found that βcat-LOF represses the activity of the Wnt reporter in the otocyst, confirming that the C-terminal domain of β-catenin is required for its transcriptional activity (*Herrera et al., 2014*). Therefore, some of the effects reported in Stevens et al. (ectopic and fused vestibular sensory organs) after truncated β-catenin overexpression can be reconciled with our results and those of *Rakowiecki and Epstein, 2013* if one considers that these were elicited by a reduction, and not a gain, of Wnt activity. One

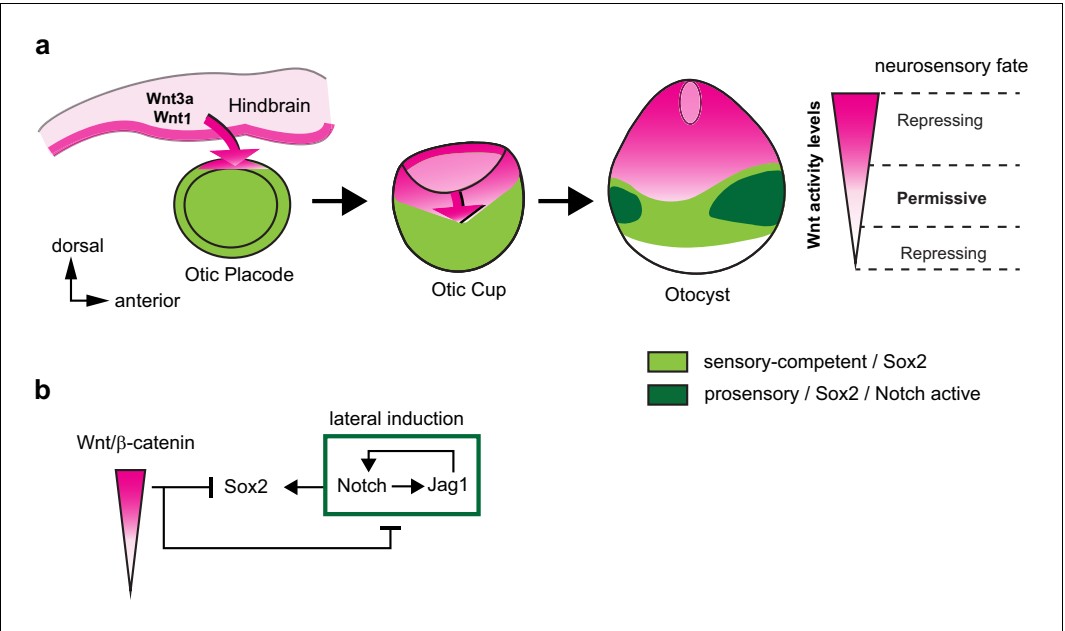

**Figure 7.** A schematic model of the effects of canonical Wnt activity on the patterning of inner ear neurosensory-competent domains. (a) The hindbrain produces Wnt1 and Wnt3a ligands activating Wnt signalling in the dorsal aspect of the otic placode. Over time, a dorso-ventral gradient of Wnt activity forms in the otic cup and otocyst and regulates in a dose-dependent manner neural and prosensory specification. At intermediate levels, Wnt activity is permissive for the maintenance of Sox2 expression and Jag1/Notch signalling, which reinforces Sox2 expression and promotes acquisition of a prosensory fate by lateral induction. Hence, the dorso-ventral gradient of Wnt activity confines Sox2 expression to a middle region of neurosensory-competence from where the individual sensory organs will originate. (b) Schematic representation of the hypothetical regulatory interactions between Wnt and Notch signalling and their impact on Sox2 expression. The connectors do not imply direct interactions and intermediary factors are likely to contribute to the feedback loops.

remaining puzzle is that ectopic vestibular-like sensory patches were present in the basilar papilla after infection with RCAS-βcat-LOF (*Stevens et al., 2003*), whilst we found that Tol2-mediated βcat-LOF overexpression completely abolishes sensory cell formation, including in the auditory organ. This discrepancy may be due to differences in the onset or levels of βcat-LOF expression after RCAS infection versus Tol2 electroporation, although further studies with inducible LOF and GOF forms of β-catenin will be necessary to confirm this and to determine if the dosage or timing of Wnt activity has an influence on specification of vestibular versus auditory organs.

## Canonical Wnt acts upstream of notch signalling during neurosensory specification

Functional interactions between Notch and Wnt signalling have been well documented during hair cell formation and otic placode formation (reviewed in *Żak et al., 2015*) but not during prosensory specification. Notch signalling functions in two different ways in the early otocyst (reviewed in *Daudet and Żak, 2020*): it regulates neuroblast formation by lateral inhibition (mediated by the ligand Dll1) and it promotes prosensory specification by lateral induction (Jag1). In this study, we found that Wnt signalling acts upstream of Notch signalling in the otic vesicle: βcat-LOF induced Jag1 expression and activation of a Hes5/Notch reporter throughout the otocyst, whilst the βcat-GOF had an opposite effect. On the other hand, forcing Notch activity by overexpressing NICD1 had no effect on the pattern of activation of the Wnt reporter in the otocyst. Nevertheless, the ability of βcat-LOF to induce large ectopic sensory territories requires Notch activity: in samples co-electroporated with βcat-LOF and DN-MAML1, which prevents the expression of Notch target genes, very few cells expressed Sox2 ectopically in the dorsal otocyst. Altogether, these results suggest that Sox2 expression is maintained by a positive feedback loop dependent on Jag1/Notch signalling (lateral induction) and repressed by a dose-dependent negative feedback from Wnt signalling

(*Figure 7b*). The interplay of long-range inhibitory (Wnt in this case) and short-range activating (such as Notch) signalling has been well studied in theoretical models of tissue patterning (*Gierer and Meinhardt, 1972*). If a short-range activator can feedback positively on its long-range inhibitor, a periodic pattern of domains of two different types, forming for example stripes, can spontaneously emerge. In the otic vesicle, Notch activity does not feedback on Wnt signalling, which could explain the initial pattern of neurosensory competence: a broad domain of Sox2-positive cells located at some distance from the long-range inhibitory signal. However, it is possible that Wnt and Notch signalling cross-interact at subsequent developmental stages, and such interactions may contribute to the segregation of the original 'pan-sensory' domain into multiple sensory organs. Further insights into these interactions and the dynamics of production, diffusion, and degradation of Wnt ligands will be needed to elucidate their exact morphogenetic roles throughout otic development.

## Context- and dose-dependent effects of canonical Wnt signalling in the developing inner ear

Our findings provide further evidence for the great variety of context-dependent functions of Wnt signalling during inner ear development. At early stages of inner ear development, Wnt signalling regulates otic induction (*Ladher et al., 2000*), promotes otic versus epidermal fate in the cranial ectoderm (*Ohyama et al., 2006*; *Freter et al., 2008*), and is required for vestibular system morphogenesis (*Riccomagno et al., 2005*; *Noda et al., 2012*; *Rakowiecki and Epstein, 2013*). Previous studies have shown that the overexpression of an active form of β-catenin can supress the expression of neurogenic markers in the mouse inner ear (*Ohyama et al., 2006*; *Freyer and Morrow, 2010*), suggesting that high levels of Wnt activity repress otic neurogenesis. Our loss-of-function experiments and RNA-Seq analysis confirm this and point at a broader role for canonical Wnt as a negative regulator of both neuronal and prosensory specification in the otic vesicle. At later stages, however, Wnt activity becomes elevated in prosensory domains and has been implicated in the control of progenitor cell proliferation (*Jacques et al., 2012*; *Jacques et al., 2014*) and the patterning of auditory epithelia (*Munnamalai and Fekete, 2016*; *Munnamalai et al., 2017*). These context-specific roles could be explained by distinct co-factors or epigenetic changes that could modify the identify of Wnt target genes in different cell types and at different developmental stages. Another important factor, highlighted by our findings, is the dosage of Wnt activity: otic progenitors must be exposed to intermediate levels of Wnt activity to maintain a neurosensory fate and sustained activation of Wnt signalling may lead to cell death in the early otocyst. These insights are directly relevant to the design of improved protocols for the derivation of inner ear organoids from embryonic stem cells. In fact, our results could explain the effects of the Wnt agonist CHIR99021 (CHIR) on 3D stem-cell derived inner ear organoids: intermediate doses of CHIR promote sensory cell formation, but high doses reduce it (*DeJonge et al., 2016*). In their study, the authors used CHIR at a relatively early stage of organoid formation and concluded that the improvement with intermediate doses of CHIR was due to the ability of Wnt activity to promote otic induction (*DeJonge et al., 2016*). Our results do not refute this possibility, but they indicate that the time- and dose-dependent effects of Wnt activity on prosensory cell specification must also be considered to improve current protocols for in vitro derivation of inner ear sensory cells.

The major challenge ahead is to understand how the large repertoire of Wnt ligands, Frizzled receptors, and modulators of the Wnt pathway expressed in a dynamic manner in the embryonic inner ear (*Noda et al., 2012*; *Sienknecht and Fekete, 2009*; *Sienknecht and Fekete, 2008*) regulate both the levels and spatial patterns of Wnt activity. In addition, a membrane-tethered form of Wingless can still elicit a gradient of Wnt activity in the *Drosophila* wing disc, due to the overall growth of the epithelium itself (*Alexandre et al., 2014*). It is therefore conceivable that the growth and complex 3D remodelling of the inner ear could shape the patterns of Wnt activity during its development. Another important goal is to understand how the transcriptional targets of canonical Wnt signalling regulate otic neurosensory specification and the molecular basis of their interactions with Notch signalling. Our findings provide a new framework to explore these questions and the roles of Wnt ligands as tissue morphogens in the inner ear as well as in organoid systems.

# Materials and methods

**Key resources table**

| Reagent type (species) or resource | Designation | Source or reference | Identifiers | Additional information |
|---|---|---|---|---|
| Gene (*Mus musculus*) | B-catenin (Ctnnb1) | GenBank | | |
| Software, algorithm | R (RRID:SCR_001905) | https://www.r-project.org/ | | Used for quantification and visualisation |
| Software, algorithm | Volocity (RRID:SCR_002668) | https://quorum technologies.com/index.php/component/content/category/31-volocity-software | | Used for quantification |
| Software, algorithm | ImageJ (RRID:SCR_003070) | https://www.imagej.net | | Used for quantification and visualisation |
| Software, algorithm | OriginPro 2020 | OriginLab Corporation | | Used for statistical analysis and visualisation |
| Recombinant DNA reagent | 5TCF::H2B-RFP (plasmid) | PMID:24942669 | | Wnt reporter |
| Recombinant DNA reagent | T2-5TCF::nd2Scarlet (plasmid) | This paper and PMID:27869816 | | Wnt reporter cloned into Tol2 transposon system, Daudet lab |
| Recombinant DNA reagent | T2-Hes5::nd2EGFP (plasmid) | PMID:22991441 | | Notch reporter |
| Recombinant DNA reagent | Hes5::d2FP635 (plasmid) | PMID:22991441 | | Notch reporter |
| Recombinant DNA reagent | RCAS-βcat-LOF (plasmid) | PMID:12941626 PMID:7876319 | | β-catenin LOF |
| Recombinant DNA reagent | T2-βcat-LOF (plasmid) | This study | | β-catenin LOF cloned into Tol2 transposon system, Daudet lab |
| Recombinant DNA reagent | PB-βcat-GOF (plasmid) | PMID:24942669 | | β-catenin GOF |
| Recombinant DNA reagent | T2-βcat-GOF (plasmid) | This study | | β-catenin GOF cloned into Tol2 transposon system, Daudet lab |
| Recombinant DNA reagent | pNICD1-EGFP (plasmid) | PMID:15634704 | | Notch GOF |
| Recombinant DNA reagent | pDN-MAML1-EGFP (plasmid) | PMID:27218451 | | Notch LOF |
| Recombinant DNA reagent | T2-EGFP (plasmid) | PMID:17362912 | | Control plasmid |
| Recombinant DNA reagent | T2-mEGFP (plasmid) | This study | | Control plasmid, mEGFP cloned into Tol2 transposon system, Daudet lab |
| Recombinant DNA reagent | T2-mRFP (plasmid) | This study | | Control plasmid, mRFP cloned into Tol2 transposon system, Daudet lab |
| Recombinant DNA reagent | pTurquoise (plasmid) (RRID:Addgene_98817) | Addgene | Addgene No: 98817 | Control plasmid |
| Recombinant DNA reagent | mPB (plasmid) | PMID:19755504 | | PiggyBac transposase |
| Recombinant DNA reagent | pCAGGS-T2-TP (plasmid) | PMID:17362912 | | Tol2 Transposase |

*Continued on next page*

*Continued*

| Reagent type (species) or resource | Designation | Source or reference | Identifiers | Additional information |
|---|---|---|---|---|
| Commercial assay or kit | In-Fusion HD Cloning | Takarabio | No: 638916 | |
| Commercial assay or kit | RNAqueous-Micro Total RNA Isolation Kit | Life Technologies | No: AM1931 | |
| Antibody | Rabbit polyclonal anti-Jagged 1 (RRID:AB_649685) | Santa-Cruz Biotechnology | No: sc-8303 | IF (1:200) |
| Antibody | Rabbit polyclonal anti-Sox2 (RRID:AB_2341193) | Abcam | No: 97959 | IF (1:500) |
| Antibody | Mouse IgG1 monoclonal anti-Sox2 (RRID:AB_10694256) | BD Biosciences | No: 561469 | IF (1:500) |
| Antibody | Mouse monoclonal IgG1 anti-Islet1 (RRID:AB_1157901) | Developmental Studies Hybridoma Bank | Clone 39.3F7 | IF (1:250) |
| Antibody | Mouse monoclonal IgG1 anti-HA-tag (RRID:AB_291262) | Babco Inc | No: MMS-101R | IF (1:500) |
| Antibody | Mouse monoclonal IgG1 anti-Myo7a (RRID:AB_2282417) | Developmental Studies Hybridoma Bank | Clone 138–1 | IF (1:500) |
| Antibody | Mouse monoclonal IgG1 anti-HCA (RRID:AB_2314626) | Guy Richardson | | IF (1:1000) |
| Chemical compound, drug | LiCl | Sigma-Aldrich | No: L7026 | Concentrations: 5 μM, 15 μM, 25 μM, 35 μM |
| Chemical compound, drug | IWR-1 | Sigma-Aldrich | No: I0161 | Concentration 300 μM |
| Chemical compound, drug | Leibovitz's | Gibco | No: 21083–027 | |
| Chemical compound, drug | Matrigel | Corning | No: 354230 | |
| Chemical compound, drug | DMEM/F12 | Gibco | No: 21041–025 | |
| Chemical compound, drug | HEPES | Sigma-Aldrich | No: SRE 0065 | Concentration 1% |
| Chemical compound, drug | Ciprofloxacin | Fluka | No: 17850–5 G-F | Concentration 0.1% |
| Sequence-based reagent | 5xTCF-BS_F | This paper | PCR primers | ATGGGCCCTCGTCGAACGACGTTGTAAAACGACGG |
| Sequence-based reagent | 5xTCF-BS_R | This paper | PCR primers | TGGTGGCgAGATCTGCGGCACGCTG |
| Sequence-based reagent | Bcat_GOF_F | This paper | PCR primers | TTTTGGCAAAGAATTGCCACCATGGCTACTCAAGC |
| Sequence-based reagent | Bcat_GOF_R | This paper | PCR primers | TAGACTCGAGGAATTtcacctattatcacggccgcc |
| Sequence-based reagent | Bcat_LOF_F | This paper | PCR primers | gattacgctgctcgagcaatccccgagc |
| Sequence-based reagent | Bcat_LOF_R | This paper | PCR primers | ctagagtgaagcagctcagtaagag |

## Animals

Fertilised White Leghorn chicken (*Gallus gallus*) eggs were obtained from Henry Stewart UK and incubated at 37.8°C for the designated times. Embryonic stages refer to embryonic days (E), with E1 corresponding to 24 hr of incubation or to *Hamburger and Hamilton, 1992* stages. Embryos older than E5 were sacrificed by decapitation. All procedures were approved by University College London local Ethics Committee.

## In ovo electroporation

Electroporation (EP) of the otic placode/cup of E2 chick embryos (stage HH 10–14) was performed using a BTX ECM 830 Electro Square Porator as previously described (*Freeman et al., 2012*). The total concentration of plasmid DNA ranged for each set of experiments between 0.5 and 1 µg/µl. Unless otherwise specified, a minimum number of six successfully transfected samples were examined for each experimental condition.

## Plasmids

The plasmids used in this study and their origin are described in the Appendix 1 file. New constructs were generated by standard subcloning methods or using the In-Fusion HD Cloning Kit (Takarabio). All plasmids used for in ovo electroporation were purified using the Qiagen Plasmid Plus Midi Kit (Qiagen).

## Wnt gradient quantification

Chicken embryos were electroporated at E2 with 5TCF::H2B-RFP (Wnt reporter) and T2-EGFP (control) plasmids. Otocysts were collected 24 hr post-electroporation and confocal stacks (16-bit pixel intensity scale) were taken from whole mount preparations. Seven almost fully transfected otocysts (based on EGFP expression) were selected for further analyses using the Volocity software (RRID: SCR_002668). The EGFP channel was used to outline the otocyst region of interest (ROI). Next, the commands 'Finding Object' and 'Filter Population' (same settings for each otocyst) were applied to the RFP channel to detect cell nuclei positive for Wnt reporter within the ROI. The 'Separate Touching Objects' function was used to segment individual cell nuclei. Mean RFP fluorescence intensity values and X,Y coordinates of individual nuclei were exported to Excel, normalised by Min–Max scaling for each individual otocyst and plotted using ggplot2 in R (RRID:SCR_001905). To analyse the profile of the Wnt gradient, the median and standard deviation of RFP intensity of groups of nuclei were calculated in 10% increment steps along the dorso-ventral (Y) axis of the otocyst and plotted using ggplot2 (see also Appendix 1).

## Immunohistochemistry

Entire chicken embryos (E3–E4) or their heads (>E5) were collected, fixed for 1.5–2 hr in 4% paraformaldehyde (PFA) in 0.1 M phosphate buffered saline (PBS), and processed for whole-mount immunostaining using conventional methods. Further details of the protocol and reagents can be found in the Appendix 1 file. The following antibodies were used: rabbit anti-Jagged 1 (Santa-Cruz Biotechnology, Dallas, TX; sc-8303; 1:200), rabbit anti-Sox2 (Abcam, UK; 97959, 1:500), mouse IgG1 monoclonal anti-Sox2 (BD Biosciences, San Jose, CA; 561469, 1:500), mouse IgG1 anti-Islet1 (Developmental Studies Hybridoma Bank, Iowa City, IA; Clone 39.3F7, 1:250), mouse IgG1 anti-HA-tag (Babco Inc, Richmond, CA; MMS-101R, 1:500), mouse IgG1 anti-Myo7a (Developmental Studies Hybridoma Bank, 1:500), and mouse IgG1 anti-HCA (a kind gift of Guy Richardson, 1:1000). Secondary goat antibodies conjugated to Alexa dyes (1:1000) were obtained from Thermo Fischer Scientific (UK). Confocal stacks were acquired using a Zeiss LSM880 inverted confocal microscope and further processed with ImageJ.

## Quantification of Sox2 expression

Confocal stacks (12 bits) of samples transfected with T2-βcat-GOF were analysed using the ImageJ Time Series Analyzer plugin (J. Balaji 2007; Dept. of Neurobiology, UCLA). After background correction of the images (each a single confocal Z-plane), the average levels of Sox2 and GFP fluorescence were measured in manually selected prosensory cell nuclei using a 4 µm diameter circle selection tool. The measurements from two to three optical slices from the same confocal stack were combined and analysed using the OriginPro software.

## Organotypic cultures

Dissections were performed in ice-cold L-15 medium (Leibovitz). E3 embryos were halved along the midline, the head and trunk were removed; the otocysts with surrounding region including the hindbrain were placed in 35 mm Mattek dishes coated with a thick layer of ice-cold Matrigel (Corning). Next, samples were incubated in a culture incubator (5% $CO_2$, 37°C) for 30 min to allow

polymerisation of Matrigel. Samples were then incubated for 24 hr in approximately 250–300 µl of DMEM/F12 medium with Phenol Red (Invitrogen) containing 1% HEPES, 0.1% CIPRO, and supplemented with LiCl, IWR-1, or vehicle at matched concentration in control experiments. On the next day, samples were washed in ice-cold PBS, fixed for 1.5 hr in PBS containing 4% PFA, and processed for immunohistochemistry. Otocysts electroporated with 5TCF:H2B-RFP were cultured for 24 hr in medium supplemented with LiCl and IWR-1 to assess their effects on Wnt activity. The working concentration of IWR-1 (300 µM) was determined by qPCR (see Appendix 1).

## Acknowledgements

We thank Thea Støle and Caitlin Broadbent (UCL Ear Institute) and Paola Niola and Tony Brooks (UCL Genomics) for excellent technical support. We are grateful to the following researchers for sharing essential plasmids with us: Sebastian Pons (5TCF::H2B-RFP, PB-βcat-GOF, mPB), Donna Fekete (RCAS-βcat-LOF), and Christophe Marcelle (pDN-MAML1-EGFP). We thank Guy Richardson for the generous gift of the HCA antibody.

The mouse IgG1 anti-Islet1 antibody (clone 39.3F7) developed by TM Jessell and S Brenner-Morton was obtained from the Developmental Studies Hybridoma Bank, created by the NICHD of the NIH and maintained at The University of Iowa, Department of Biology, Iowa City, IA 52242. This work was supported by an Action on Hearing Loss International Research Grant (G76; MZ) and the Medical Research Council (MR/S003029/1; ND). We thank Donna Fekete, Andy Forge, and Jonathan Gale for their comments on the manuscript.

## Additional information

### Funding

| Funder | Grant reference number | Author |
| --- | --- | --- |
| Medical Research Council | MR/S003029/1 | Nicolas Daudet |
| Action on Hearing Loss | G76 | Magdalena Żak<br>Nicolas Daudet |

The funders had no role in study design, data collection and interpretation, or the decision to submit the work for publication.

### Author contributions

Magdalena Żak, Conceptualization, Formal analysis, Funding acquisition, Investigation, Visualization, Methodology, Writing - original draft, Writing - review and editing; Nicolas Daudet, Conceptualization, Supervision, Funding acquisition, Investigation, Methodology, Writing - original draft, Project administration, Writing - review and editing

### Author ORCIDs

Nicolas Daudet ⓘ https://orcid.org/0000-0002-4039-4716

### Ethics

Animal experimentation: All experimental procedures on fertilized chicken eggs (2–8 days of incubation) were carried out in accordance with the United Kingdom Animals (Scientific Procedures) Act (ASPA) of 1986 and following the "3Rs" principles (Replacement, Reduction and Refinement) in conducting animal research. As per the ASPA 1986, the use of chicken embryos (*Gallus gallus*) aged less than two thirds of the incubation period does not require formal approval and a Home Office Project Licence.

### Decision letter and Author response

Decision letter https://doi.org/10.7554/eLife.59540.sa1
Author response https://doi.org/10.7554/eLife.59540.sa2

## Additional files

### Supplementary files
• Transparent reporting form

### Data availability
Source data files have been provided for the quantification of the Wnt reporter shown in Figure 1.

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

# Appendix 1

## Supplementary materials and methods
Plasmids

Plasmids used and generated for the study are listed in the table below.

| Plasmid name | Plasmid type | Insert | Promoter | References |
|---|---|---|---|---|
| 5TCF::H2B-RFP | Wnt reporter (TopRed) | Five x TCF/Lef binding sites; H2B-RFP fusion protein | Minimal TK | 5xTCF-bs-RFP (*Herrera et al., 2014*) |
| T2-5TCF::nd2Scarlet | Wnt reporter (Tol2) | Five x TCF/Lef binding sites; nuclear-localised and destabilised Scarlet | Minimal TK | This study and *Bindels et al., 2017* |
| T2-Hes5::nd2EGFP | Notch reporter (Tol2) | Mouse Hes5 promoter; nuclear-localised and destabilised EGFP | Mouse Hes5 | *Chrysostomou et al., 2012* |
| Hes5::d2FP635 | Notch reporter (Slax) | Mouse Hes5 promoter; destabilised Turbo FP635 | Mouse Hes5 | *Chrysostomou et al., 2012* |
| RCAS-βcat-LOF | β-catenin LOF (RCAS) | HA-tagged truncated form of *Xenopus* β-catenin (lacking 134aa in C-terminus and 147aa at the N-terminus) | LTR | RCAS/*β-catenin (*Stevens et al., 2003*) Truncated *Xenopus* β-catenin « T5 » construct in *Funayama et al., 1995* |
| T2-βcat-LOF | β-catenin LOF (Tol2) | Membrane-localised Cherry; 2A self-cleaving peptide; triple HA-tagged truncated form of *Xenopus* β-catenin (subcloned from RCAS-βcat-LOF) | CAG | This study |
| PB-βcat-GOF | β-catenin GOF (PiggyBac) | S33Y* mutated form of human β-catenin; IRES; H2B-EGFP fusion protein | CAG | PiggyBac-CAG-β-catenin$^{S33Y}$-IRES-EGFP (*Herrera et al., 2014*) |
| T2-βcat-GOF | β-catenin GOF (Tol2) | S33Y* mutated form of human β-catenin; IRES; H2B-EGFP fusion protein | CAG | This study |
| pNICD1-EGFP | Notch GOF (pCAGGS) | HA-tagged chicken Notch1 intracellular domain; IRES; EGFP | CAG | NICD1-IRES-GFP (*Daudet and Lewis, 2005*) |
| pDN-MAML1-EGFP | Notch LOF (pCAGGS) | Truncated, dominant-negative form of human Mastermind-like one fused to EGFP | CAG | CAGGS-DN-MAML1–EGFP (*Sieiro et al., 2016*) |
| T2-EGFP | Control (Tol2) | Enhanced green fluorescent protein | CAG | pT2K-CAGGS-EGFP (*Sato et al., 2007*) |
| T2-mEGFP | Control (Tol2) | Membrane-localised enhanced green fluorescent protein | CAG | This study |
| T2-mRFP | Control (Tol2) | Membrane-localised cherry | CAG | This study |
| 3xnls-mTurquoise2 | Control (pCAGGS) | Nuclear-localised turquoise | CAG | 3xnls-mTurquoise2 (*Chertkova AO et al., 2017*) |
| mPB | PiggyBac transposase (pCAGGS) | PiggyBac transposase | CAG | *Lu et al., 2009* |
| pCAGGS-T2-TP | Tol2 transposase (pCAGGS) | Tol2 transposase | CAG | *Sato et al., 2007* |

## Quantification of the Wnt gradient profile

A schematic representing the pipeline for analysis of fluorescence intensity profile of Wnt reporter along the dorso-ventral axis of otocyst transfected with 5TCF::H2B-RFP is shown in *Appendix 1—figure 1*.

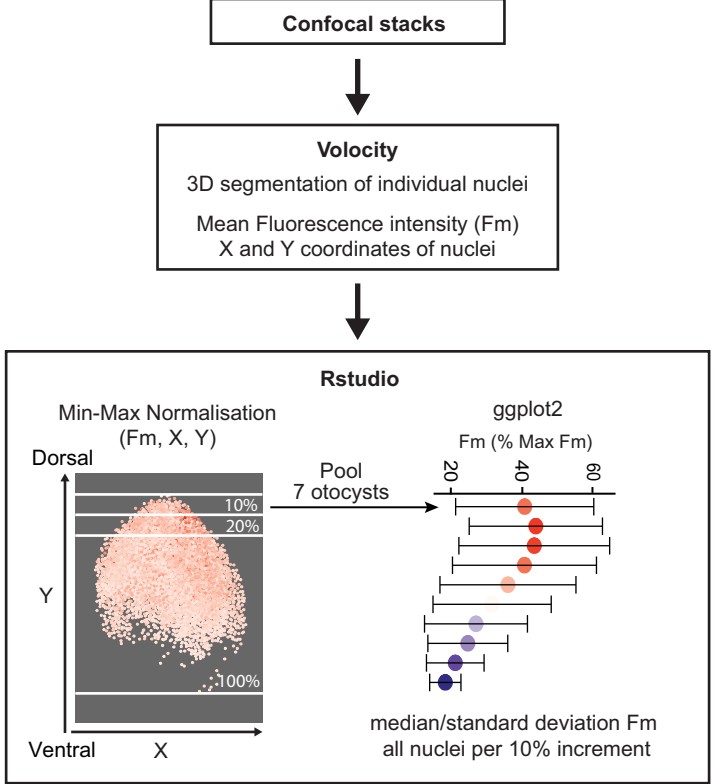

**Appendix 1—figure 1.** Quantification of the Wnt gradient profile.

## Immunohistochemistry

Sample dissection protocol depended on the age of embryos. For E7 embryos, the heads were halved along the midline, the inner ear was dissected, and the otic cartilage was removed from the basilar papilla and trimmed at the dorsal side to expose the otic epithelium. For embryos aged E3–E4, the embryo was dissected along the midline, the hindbrain was removed, and the region surrounding the otocyst was only partially trimmed to facilitate orientation. A small opening was made at the dorsal tip of the otocyst using a fine needle and the tissue was permeabilised in PBS containing 0.3% Triton and 10% goat serum for 30 min at room temperature. Specimens were incubated with primary antibodies diluted in 0.1% Triton in PBS at 4°C overnight. On the next day, tissues were rinsed with PBS at room temperature and incubated with secondary antibodies diluted in 0.1% Triton and 10% goat serum at 4°C overnight. Afterward, tissues were again rinsed with PBS and mounted in Vectashield Antifade Mounting Medium (Vector laboratories). A fine layer of vacuum grease was applied between the slide and coverslip to avoid excessive flattening of the tissue.

## Quantification of fluorescence intensity levels of the reporters

Confocal stacks (16-bit pixel intensity scale) of three otocysts electroporated with 5TCF::H2B-RFP (Wnt reporter) and T2-Hes5::nd2EGFP (Notch reporter) were analysed using the Volocity software and the protocol described for Wnt gradient quantification. Mean fluorescent intensity values of both RFP and EGFP channels were obtained for each reporter and plotted using ggplot2 in RStudio.

## Quantification of Sox2 expression in βcat-GOF transfected samples

Confocal stacks (12-bit images) obtained from βcat-GOF transfected samples immunostained for Sox2 expression were analysed using the ImageJ plot profile function (RRID:SCR_003070). A free-hand straight line was drawn across the transfected region and the fluorescent intensity profile for the Sox2 and EGFP channels was generated. The results were plotted using ggplot2 in RStudio. Two confocal stacks (12-bit intensity scale) were analysed using the ImageJ Time Series Analyzer plugin (J. Balaji 2007; Dept. of Neurobiology, UCLA). After background correction of the images (each a single confocal Z-plane), the average levels of Sox2 and GFP fluorescence were measured in manually selected prosensory cell nuclei using a 4 µm diameter circle selection tool. The measurements from two to three optical slices from the same confocal stack were combined and analysed using the OriginPro software.

Determination of IWR-1 working concentration qPCR was used to establish a working concentration of IWR-1 in organotypic culture. E3 Otic explants (four to five chicken otocysts per condition) were incubated in media containing 50 µM, 150 µM, and 300 µM of IWR-1 or DMSO (vehicle) as a control. After 24 hr incubation, total RNA was isolated using the RNAqueous-Micro Total RNA Isolation Kit (Ambion) and reverse transcribed using iScript cDNA Synthesis Kit (Bio-Rad). qPCR reactions were performed with Quantifast Syber Green (Qiagen). The effects of the treatment were analysed by testing the decrease in the expression levels of Lgr5 and Axin2, two genes positively regulated by Wnt signalling (*Barker et al., 2007*; *Yan et al., 2001*). The relative quantification of expression was analysed using the ΔΔCt method (*Pfaffl, 2001*) and showed the significant downregulation, 55% for Lgr5% and 78% for Axin2, at 300 µM of IWR-1.

