## [Decision Letter]

**Acceptance summary:**

The dorsal-ventral axis of the inner ear is established by factors secreted from the hindbrain with Wnts emanating from the dorsal and Sonic hedgehog from the ventral region. Using gain and loss of function approaches in the developing chicken inner ear, this study demonstrated that despite the source of Wnts is from the dorsal hindbrain, Wnt has a dose-response effect on the developing inner ear. A high level of Wnt signaling is required in the dorsal otic region to suppress the neural-sensory fate, whereas a lower level in the ventral otic region is required for specifying the neural-sensory fate. These results advance the understanding of the role of Wnts in patterning the dorsal-ventral axis of the inner ear.

**Decision letter after peer review:**

Thank you for submitting your article "A gradient of Wnt activity positions the neurosensory domains of the inner ear" for consideration by *eLife*. Your article has been reviewed by Kathryn Cheah as the Senior Editor, a Reviewing Editor, and three reviewers. The reviewers have opted to remain anonymous.

The reviewers have discussed the reviews with one another and the Reviewing Editor has drafted this decision to help you prepare a revised submission.

This study provided evidence that a dorsal to ventral gradient of Wnt signaling in the developing chicken inner ear mediates the position of the neurosensory domain. The co-electroporation of reporter and activation constructs into the otic epithelium allowed simultaneously direct read outs and modulation of Wnt signaling. High levels of Wnt activity repress neurosensory specification and Notch pathway but Notch activity has no effect on Wnt signaling. On the other hand, blocking Wnt activity induced ectopic neurosensory domain dorsally, whereas it repressed prosensory specification ventrally.

The reviewers all appreciated the approach and execution of the study. However, there are many outstanding issues regarding the quantification and interpretation of the results that require additional experiments.

Essential revisions:

1) Quantification and validation of results

a) Most results are based on individual images without quantification. In some places, the conclusions are not supported by the images, for instance in Figure 3B/C it is concluded that expression of βcat-GOF blocks *Sox2* expression. While the arrows indicate two regions where this is true, there are clearly other close by cells expressing the βcat-GOF but also still positive for *Sox2*. Additional examples with more extensive transgene expression would be helpful in providing the correlation between transgene expression and change in prosensory expression.

b) The results of the RNAseq experiments are confusing. First of all, it is incorrect to state that something was increased but not significantly. This is the point of statistics to be able to identify real changes from changes that are within the variation of the results. Non-significant increases are not real increases. So, neither *Sox2* nor Jag1 was increased despite the images shown in Figure 6A? This seems hard to believe and suggests a possible technical issue. Consistent with that, the separation between control samples 3 and samples 2 and 1 in the PCA suggests a technical issue with the collection of sample 3.

Overall, the RNAseq results yielded little insight into the prosensory issue, although did suggest a neuronal issue, although that wasn't fully explored (see below).

c) Several proneural bHLHs are identified as up-regulated in the RNAseq but with the exception of Neurod1, are any of these expressed in chick auditory neurons? Also, Islet1 is then used to demonstrate new neuronal formation. Is Islet1 increased in the RNAseq results?

e) Subsection “Canonical Wnt activity forms a dorso-ventral gradient in the otocyst and is reduced in neurogenic and prosensory domains”: there appears to be no transfection control for the wnt/notch study, correct?

f) Figure 1E,F-F’: Was any quantification done to support the inverse relationship concept?

2) Additional supporting evidence

a) Figure 2D,D’: the EGFP expression appears to be restricted to the anterior ventral region of the otocyst but wnt signaling has disappeared throughout the otocyst. Does this make sense?

b) Figure 4B/C if all the cells that expressed βcat-GOF are gone by E7, isn't that a potentially significant amount of cell death that could influence the overall formation of the otocyst?

c) It is difficult to make clear conclusions from the longer-term experiments examining sensory cell formation (Figure 4). In the GOF experiments there is loss of the transgene-expressing cells which may explain the loss of sensory cells. In the LOF experiments while there may be additional sensory regions dorsally the ear is so malformed it is difficult to make that determination. In the LOF experiments, while the transgene seems fairly uniformly expressed, the sensory regions seem broken into smaller, more frequent domains, although the reason for this is unclear. It would be important to understand the morphology changes in these ears.

d) In Figure 6, using the IWR-1 Wnt inhibitor it would be expected that the most ventral domain would be devoid of prosensory expression (as it was proposed that low levels of Wnt expression is required for prosensory expression), yet it seems like more of a medial domain that is devoid of prosensory expression, is this consistent?

e) The finding of increased neurogenesis is interesting (Figure 7) but not well investigated. It is an outstanding question in the field as to why neurogenesis only occurs in the anterior prosensory portion of the otocyst, and not the posterior portion. These data suggest Wnt signaling may play a role and that decreased Wnt signaling leads to neurogenesis in the posterior regions as well. This would be consistent with the gradient of normal Wnt that is not as extensive in the anterior region (Figure 1C). Is there decreased neurogenesis in the GOF? Additional markers and neurogenesis analysis in other manipulations (such as GOF and IWR cultures) would better reveal the effects of Wnt signaling on neuro-competence, as well as some discussion.

f) The model (Figure 8) suggests that Wnt could act directly on *Sox2*, but could also act through Notch, which is known to regulate *Sox2*. The authors clearly show that Wnt can affect Notch expression, and that in the absence of Notch there is no expansion of the prosensory domain (Figure 3—figure supplement 2). Moreover, in some cases, the effects on Jag1/Notch are more dramatic that on *Sox2* (eg Figure 3). Is there any evidence that Wnt could directly act on *Sox2*? It seems likely that Wnt is acting indirectly to regulate sensory formation via Notch. Although there is no indication of this in the model, or in the discussion.

3) Points to consider

a) What is the overarching picture on the role of Wnt: A high level of Wnt acts acts as a dorsalizing signal, but is it capable of specifying vestibular sensory patches as well, or is a certain level of Wnt signaling required for any prosensory formation? Is it specific to the auditory organ only?

b) In Munnamalai et al., (2017), Wnt9a- over expression in the chicken ototcyst elicited some vestibular patches in the basilar papilla. The data presented by the authors here would suggest that somehow Wnt signaling was first suppressive to allow vestibular fates, before allowing being downregulated to allow a prosensory fate. But since it was already committed to behave as vestibular, it adopted a vestibular sensory fate. To align all these studies, this would suggest some kind of negative feed-back on the Wnt pathway itself is required to elicit vestibular patches within the basilar papilla as described.

Following this thought, the authors show in Figure 4B, that T2-βcat-GOF overexpression shows no active Wnt signaling by E7. The authors suggest that those cells are “eliminated”. However, I would offer an alternative explanation that by E7, the Wnt pathway was actively down-regulated through negative feed-back. I would like to see one additional experiment where T2-βcat-GOF is over expressed and probed for Wnt-regulated, Wnt-inhibitors such as Axin2/ Nkd1 (PMCID-PMC4462952).

c) Are these ectopic patches (Figure 4) “vestibular” sensory patches? Use a vestibular marker such as gm2. This would help address if an intermediate level of Wnt is necessary for the formation of any prosensory epithelia? Or is it specific to cochlear sensory epithelia?

[Editors' note: further revisions were suggested prior to acceptance, as described below.]

Thank you for submitting your article "A gradient of Wnt activity positions the neurosensory domains of the inner ear" for consideration by *eLife*. Your article has been reviewed by Kathryn Cheah as the Senior Editor, a Reviewing Editor, and three reviewers.

The reviewers have discussed the reviews with one another and the Reviewing Editor has drafted this decision to help you prepare a revised submission.

Summary:

The developing inner ear acquires its primary axial information from the surrounding tissues early on during development. Using gain and loss of function approaches in the presence of reporters, the authors demonstrated that a dorsal-ventral (D-V) gradient of Wnt activity provides the positional information for the neurosensory domain to form in the ventral otocyst.

Essential revisions:

1) Many questions were raised by the reviewers on the RNAseq results. After a thorough discussion, we recommend that the RNAseq data be removed from the current manuscript unless the authors could address the specific comments listed below.

– The variability in Dataset 2 seems larger than that on Dataset 1 and thus appears to be weaker than dataset 1. Neither provide a Wnt target candidate that regulates otic sensory specification (i.e. Jag1-mediated prosensory specification). Only 18 genes were shared between the two datasets. Technically only one candidate gene is required, but the authors don't offer a candidate. This makes me less enthusiastic about it. (Neuroblast delamination is a separate phenomenon altogether). Isl1 is also expressed in the epithelium, so is it a potential candidate for prosensory specification?

– The RNA-seq results failed to show significant changes in either Jag1 or *Sox2*, despite clear changes in the immunohistochemistry (Figure 7). To address this the authors did an additional RNA-seq analysis, in this case using 5 treated otocysts and controls instead of 3. In this dataset they still failed to detect differences in Jag1 or *Sox2*, or other known sensory genes. Moreover, there was very little overlap in gene changes between the 2 datasets. Thus this additional dataset failed to resolve the issues in the original dataset, despite having more samples, and does not yield any additional insights into the role of Wnt signaling on prosensory development, although did confirm potential changes in neuronal development.

– While it is commendable that the RNAseq study was repeated, the lack of consistency in the results raises, for me, serious concerns about reliability. In particular considering that the work was done in the same lab using the same techniques, one would expect to see more consistent results.

2) Provide a better quantification of the efficiency of downregulation of *Sox2* by GOF Wnt activity.

– The authors addressed the concern of lack of quantification by measuring fluorescence levels in two regions in two different otocysts, showing (in most cases in these chosen regions) an inverse correlation between transfected Wnt activity and *Sox2* expression, as proposed. This correlation is interesting but does not address the concern that there were regions with GFP expression (reflecting high Wnt activity) and no apparent downregulation of *Sox2*. They acknowledged that there may be transgene expression differences. They changed the text to: The overexpression of βcat-GOF reduced, in a cell-autonomous manner, the levels of Jag1 and *Sox2* expression in the majority of transfected prosensory cells but did not induce any change in the dorsal region of the otocyst (n=6)." However it is still unclear whether the Βcat-GOF decreased *Sox2* in the majority of GFP-positive regions, since this quantification was not done.

---

## [Author Response]

This study provided evidence that a dorsal to ventral gradient of Wnt signaling in the developing chicken inner ear mediates the position of the neurosensory domain. The co-electroporation of reporter and activation constructs into the otic epithelium allowed simultaneously direct read outs and modulation of Wnt signaling. High levels of Wnt activity repress neurosensory specification and Notch pathway but Notch activity has no effect on Wnt signaling. On the other hand, blocking Wnt activity induced ectopic neurosensory domain dorsally, whereas it repressed prosensory specification ventrally.The reviewers all appreciated the approach and execution of the study. However, there are many outstanding issues regarding the quantification and interpretation of the results that require additional experiments.Essential revisions:1) Quantification and validation of resultsa) Most results are based on individual images without quantification. In some places, the conclusions are not supported by the images, for instance in Figure 3B/C it is concluded that expression of βcat-GOF blocks Sox2 expression. While the arrows indicate two regions where this is true, there are clearly other close by cells expressing the βcat-GOF but also still positive for Sox2. Additional examples with more extensive transgene expression would be helpful in providing the correlation between transgene expression and change in prosensory expression.

Using the ImageJ plot profile tool, we have now quantified the fluorescence intensity levels of EGFP (βcat-GOF) and *Sox2* expression across transfected regions in two different otocysts. The results of these measurements are presented in Figure 3—figure supplement 1A-D. The plots revealed that the majority of cells with high EGFP fluorescence have lower levels of *Sox2* expression than untransfected cells, confirming our original conclusion. We acknowledge the fact that some of the βcat-GOF transfected cells maintain *Sox2* expression. Among possible explanations, this could be due to some variability in transgene expression and the fact that the effects of Wnt activity on *Sox2* expression are dose-dependent, as proposed by our model. We have modified the text describing these results as follows: “The overexpression of βcat-GOF reduced, in a cell-autonomous manner, the levels of Jag1 and *Sox2* expression in the majority of transfected prosensory cells but did not induce any change in the dorsal region of the otocyst (n=6) (Figure 3B-C’’, Figure 3—figure supplement 1A-D). ”

b) The results of the RNAseq experiments are confusing. First of all, it is incorrect to state that something was increased but not significantly. This is the point of statistics to be able to identify real changes from changes that are within the variation of the results. Non-significant increases are not real increases. So, neither Sox2 nor Jag1 was increased despite the images shown in Figure 6A? This seems hard to believe and suggests a possible technical issue. Consistent with that, the separation between control samples 3 and samples 2 and 1 in the PCA suggests a technical issue with the collection of sample 3.Overall, the RNAseq results yielded little insight into the prosensory issue, although did suggest a neuronal issue, although that wasn't fully explored (see below).

We have corrected the text related to the “non-significant” increases in gene expression, this was an oversight. We have reanalysed our previous data and included new RNA-Seq datasets to address the reviewer’s comments. These new data are now described in the Results section, Figure 7, Figure 7—figure supplement 1A-E and the Supplementary file 1, Supplementary file 2, Supplementary file 3, Supplementary file 4 and Supplementary file 5.

The PCA analysis of the original dataset (now “dataset1”, shown in Figure 7—figure supplement 1A-B) shows that the treated and control samples separate well along the PC1, but one control sample DMSO3 seems to be an outlier in PC2. Upon closer inspection of PC2, we identified 5 genes causing the separation of DMSO3 from the other controls: ENSGALT00000012060 (HBZ), ENSGALT00000028027 (HBBR), ENSGALT00000075828 (HBA1), ENSGALT00000068173 (HBE), ENSGALT00000087509 (STMN1). Four of these genes encode different subunits of haemoglobin, suggesting a potential contamination of the DMSO3 sample with non-otic tissue. Although none of these genes are significantly regulated across conditions (nor are thought to participate to inner ear development), we recognize that this is a valid cause for concern and have therefore decided to repeat the experiment.

The second experimental dataset (“dataset2” thereafter) consists of 5 pairs of otocysts sequenced at twice as much depth to increase the power of our analysis. We used for each dataset a comparison of the groups according to their conditions (control DMSO versus IWR-1 treated) and selected differentially expressed genes with an FDR<0.2 (a more stringent method compared to the previous one based on p-value<0.05). Consequently, we had fewer genes differentially expressed (243 in dataset1; 174 in dataset 2) compared to the previous analysis of dataset1.

For both gene sets, the top “transcription factor binding site” recognized by ToppGene was Lef1, which confirms a perturbation of the Wnt signalling pathway.

There was limited consistency in terms of the individual genes significantly regulated in dataset1 versus dataset2 (approximately 10-15% overlap only). We acknowledge that this is an indication that with this experimental approach, our power to consistently detect small changes in gene expression levels is limited. Nonetheless, we found a higher degree of convergence in the results of our gene set analysis. In fact, the Toppgene analysis showed that among the top 50 Biological Functions of each dataset, 13 were shared. Among the top 15 functions, the only 3 shared terms were all related to neurogenesis and neuronal differentiation, which strengthens our previous assertion that Wnt signalling regulates neurosensory specification.

Surprisingly, there was no significant changes in Jagged1 or *Sox2* expression in any of the datasets. This result suggests that the relative levels of expression of these genes are not changed by IWR-1 treatment, although their spatial pattern of expression is, according to our immunostaining experiments. However, a number of other genes known to be expressed within the prosensory and neurogenic domains of the inner ear were present amongst the “neurogenesis” GO function gene set. We have now included a table listing these genes and references to their expression pattern in Supplementary file 4 and Supplementary file 5.

c) Several proneural bHLHs are identified as up-regulated in the RNAseq but with the exception of Neurod1, are any of these expressed in chick auditory neurons? Also, Islet1 is then used to demonstrate new neuronal formation. Is Islet1 increased in the RNAseq results?

We have repeated the experiment using 5 pairs of otocysts (dataset2) and reanalysed our previous RNAseq results (dataset1). In contrast to the previous version of the manuscript, we used comparison between “conditions” (donor embryos were not included as covariate in the analysis) and FDR<0.2 as a threshold for selecting differentially expressed genes. Using these new parameters, the gene set analysis of GO Biological Function showed an enrichment for neurogenesis-related terms. However, the Pathway analysis did not uncover a significant change in the bHLH family of proneural genes. Nevertheless, there were significant changes in several bHLH genes known to play a part in otic neurogenesis: in dataset1, Hes5.1 (fold change 1.2), Nhlh1 (fold change 0.5), Tcf3 (fold change 0.4); in dataset2, Mxd4 (fold change 0.3). Of these, the expression of Nhlh1 was previously reported in otic neurons (Jahan et al., 2010). Hes5.1 is known to be effector of Notch signalling in the developing chick inner ear during otic neurogenesis (Daudet et al., 2007).

Islet1 is a well-known marker for otic neuroblasts (Adam et al., 1998) and it can be detected by immunohistochemistry, which was necessary to confirm the correlation between βcat-LOF transfection and ectopic neurogenesis. Although Islet1 is not on the list of significantly regulated genes, its upregulation in βcat-LOF transfected cells provides a clear (and independent) confirmation that a reduction in Wnt activity promotes neurosensory specification.

e) Subsection “Canonical Wnt activity forms a dorso-ventral gradient in the otocyst and is reduced in neurogenic and prosensory domains”: there appears to be no transfection control for the wnt/notch study, correct?

This experiment was repeated. The Wnt and Notch reporters were co-transfected with a control plasmid (pTurquoise driving expression of a blue fluorescent protein) to mark the extent of electroporation. This change was incorporated into Figure 1E-E’ and in Results.

f) Figure 1E,F-F’: Was any quantification done to support the inverse relationship concept?

This statement was based on qualitative observations only. We have now used Volocity to perform a 3D quantification of the fluorescence intensity levels of the Wnt and Notch reporters in individual cells of the anterior prosensory domain (Figure 1F). The scatter plot shows that the cells with high Notch activity have no (or very low) levels of Wnt reporter fluorescence, whilst cells with high Wnt activity have no (or very low) levels of Notch activity. This quantification supports an inverse correlation between Wnt and Notch activity, but only for cells with high levels of either Wnt or Notch activity. The details of the quantification protocol are included in the Material and methods section.

2) Additional supporting evidencea) Figure 2D,D’: the EGFP expression appears to be restricted to the anterior ventral region of the otocyst but wnt signaling has disappeared throughout the otocyst. Does this make sense?

Even with a constitutive promoter, the EGFP-expressing cells can have variable levels of fluorescence, which may explain this apparent absence of transfection (in particular if imaging settings are adjusted to avoid saturation of the brightest objects– the weakly fluorescent cells can be “lost”). We now show in Figure 2D-D’ another example in which the EGFP signal of the control plasmid is detected in dorsal as well as ventral otocyst. It shows that Wnt activity is lost throughout the otocyst with the exception of the dorsal most region, where weak fluorescence of the reporter remains.

b) Figure 4B/C if all the cells that expressed βcat-GOF are gone by E7, isn't that a potentially significant amount of cell death that could influence the overall formation of the otocyst?

We have not directly assessed cell death, but it could certainly explain the absence of βcat-GOF transfected cells at E7 and contribute to the morphological defects seen in βcat-GOF transfected inner ears. We have added a comment in the Discussion related to the potential deleterious effect of sustained Wnt activity:

“Another important factor, highlighted by our findings, is the dosage of Wnt activity: otic progenitors must be exposed to intermediate levels of Wnt activity to maintain a neurosensory competent fate and sustained activation of Wnt signalling may lead to cell death in the early otocyst. “

c) It is difficult to make clear conclusions from the longer-term experiments examining sensory cell formation (Figure 4). In the GOF experiments there is loss of the transgene-expressing cells which may explain the loss of sensory cells. In the LOF experiments while there may be additional sensory regions dorsally the ear is so malformed it is difficult to make that determination. In the LOF experiments, while the transgene seems fairly uniformly expressed, the sensory regions seem broken into smaller, more frequent domains, although the reason for this is unclear. It would be important to understand the morphology changes in these ears.

The long-term consequences of manipulating Wnt activity at prosensory stages are indeed complex but we focused in this study on the specific effects on sensory organ formation. Since βcat-GOF cells disappear by E7, it was not possible to examine their (sensory or non-sensory) fate at this stage. On the other hand, the long-term consequences of βcat-LOF transfection are clear (given the persistence of mCherry expression) and in line with what we observed at E3-4: βcat-LOF cells form ectopic sensory patches dorsally, but are diverted from a sensory fate ventrally. The fact that some mCherry-positive regions do not form ectopic sensory territories could be due to differences in the levels of transfection and transgene expression (mCherry expression levels are not uniform, and ectopic territories tend to form in high-mCherry regions). It is also possible that otic cells do not respond in the same way to Wnt manipulation: some might be already committed to a non-sensory fate and refractory to Wnt manipulation at the time of transfection. It would certainly be interesting to investigate in greater detail the range of cellular effects elicited by the up or downregulation of Wnt activity at different developmental stages and within different otic cell populations, but this goes beyond the scope of this paper.

d) In Figure 6, using the IWR-1 Wnt inhibitor it would be expected that the most ventral domain would be devoid of prosensory expression (as it was proposed that low levels of Wnt expression is required for prosensory expression), yet it seems like more of a medial domain that is devoid of prosensory expression, is this consistent?

This result may seem unexpected, but it could be explained by the difference in the developmental stages at which the electroporation (E2) and organotypic cultures (E3) experiments were performed. One possibility is that at E3, most of the *Sox2*-positive cells that compose the anterior prosensory are committed to a sensory fate and insensitive to a reduction in Wnt activity levels. In contrast, cells within the medial pansensory domain (where a reduction in *Sox2* expression was noticed) may still be “labile” and could change their fate in response to molecular signals that promote or antagonise prosensory fate. We have added a sentence in the Discussion to address this point: “In IWR-1 treated otocyst, the spatial pattern of Jag1 and *Sox2* expression was only partly reduced within the ventral domain, possibly due to the fact that some of the ventral cells might already be irreversibly committed to a prosensory fate at E3.”

e) The finding of increased neurogenesis is interesting (Figure 7) but not well investigated. It is an outstanding question in the field as to why neurogenesis only occurs in the anterior prosensory portion of the otocyst, and not the posterior portion. These data suggest Wnt signaling may play a role and that decreased Wnt signaling leads to neurogenesis in the posterior regions as well. This would be consistent with the gradient of normal Wnt that is not as extensive in the anterior region (Figure 1C). Is there decreased neurogenesis in the GOF? Additional markers and neurogenesis analysis in other manipulations (such as GOF and IWR cultures) would better reveal the effects of Wnt signaling on neuro-competence, as well as some discussion.

We believe that investigating the specific role of canonical Wnt signalling in the formation of the cochleo-vestibular neurons would require extensive work (new gain and loss of function experiments with early/late neuronal markers, quantification of neurogenesis) and is beyond the scope of this study. The fact that Wnt signalling can affect neurogenesis was suggested by our RNA-seq results and confirmed by one functional experiment (βcat-LOF induces ectopic neurogenesis), leading us to conclude that canonical Wnt regulates neurosensory specification at large. We have not investigated the extent of neurogenesis in IWR-1 treated and βcat-GOF transfected samples. However, it has been previously reported by Ohyama et al., (2006) and Freyer and Morrow, (2010) that overexpression of constitutively active βcat in the mouse otic placode leads to loss of Ngn1 and NeuroD1 expression and a repression of neurogenesis (addressing partly this reviewer’s question).

We have now modified the Discussion to include this specific point, as follows:

“ At early stages of inner ear development, Wnt signalling regulates otic induction [42], promotes otic versus epidermal fate in the cranial ectoderm [43, 44] and is required for vestibular system morphogenesis [21, 22, 36]. Previous studies had shown that the overexpression of an active form of b-catenin can supress the expression of neurogenic markers in the mouse inner ear [43, 45], suggesting that high levels of Wnt activity repress otic neurogenesis. Our loss-of-function results confirm this and point at a broader role for canonical Wnt as a negative regulator of both neuronal and prosensory specification in the otic vesicle.”

f) The model (Figure 8) suggests that Wnt could act directly on Sox2, but could also act through Notch, which is known to regulate Sox2. The authors clearly show that Wnt can affect Notch expression, and that in the absence of Notch there is no expansion of the prosensory domain (Figure 3—figure supplement 2). Moreover, in some cases, the effects on Jag1/Notch are more dramatic that on Sox2 (eg Figure 3). Is there any evidence that Wnt could directly act on Sox2? It seems likely that Wnt is acting indirectly to regulate sensory formation via Notch. Although there is no indication of this in the model, or in the discussion.

Our results do not allow us to determine whether Wnt acts directly or not on *Sox2* expression, but the data shown in Figure 3—figure supplement 2 suggest that it can influence (to a limited extent) *Sox2* expression independently of Notch signalling. Furthermore, it is difficult to tell whether the effects of Wnt LOF/GOF on Notch/Jag1 are more dramatic than those on *Sox2* expression. In fact, in βcat LOF experiments the ectopic sensory patches have a larger *Sox2*-expression domain than the Jag1-expression domain. For these reasons, we prefer to leave open the possibility that Wnt could regulate prosensory specification through both *Sox2* and Notch signalling.

We have modified the Discussion to point at the specific result suggesting that Notch/lateral induction is downstream of Wnt/*Sox2*: “Nevertheless, the ability of βcat-LOF to induce large ectopic sensory territories requires Notch activity: in samples co-electroporated with βcat-LOF and DN-MAML1, which prevents the expression of Notch target genes, very few cells expressed *Sox2* ectopically in the dorsal otocyst.”

We have also modified the legend of Figure 9 (ex-Figure 8) as follows:

– “At intermediate levels, Wnt activity is permissive for the maintenance of *Sox2* expression and Jag1/Notch signalling, which reinforces *Sox2* expression and promotes acquisition of a prosensory fate by lateral induction.”

– (“b) schematic representation of the hypothetical regulatory interactions between Wnt and Notch signalling and their impact on *Sox2* expression. The connectors do not imply direct interactions and intermediary factors are likely to contribute to the feedback loops.”

3) Points to considera) What is the overarching picture on the role of Wnt: A high level of Wnt acts acts as a dorsalizing signal, but is it capable of specifying vestibular sensory patches as well, or is a certain level of Wnt signaling required for any prosensory formation? Is it specific to the auditory organ only?

The main conclusion of our study is that Wnt signalling regulates in a dose-dependent manner the spatial regulation of neurosensory specification. The gradient of Wnt activity could promote at high activity levels the specification of the dorsal-most territories of the inner ear, and be permissive for neurosensory specification at intermediate levels. It is also possible that the dosage of Wnt activity affects the type (auditory versus vestibular) of sensory organs generated, but we have not addressed this question directly in our experiments.

We have modified the first section of the Discussion to refer more specifically to this outstanding question:

“One remaining puzzle is that ectopic vestibular-like sensory patches were present in the basilar papilla after infection with RCAS-βcat-LOF [37], whilst we found that Tol2-mediated βcat-LOF overexpression completely abolishes sensory cell formation, including in the auditory organ. This discrepancy may be due to differences in the onset or levels of βcat-LOF expression after RCAS infection versus Tol2 electroporation, although further studies with inducible LOF and GOF forms of b-catenin will be necessary to confirm this and to determine if the dosage or timing of Wnt activity has an influence on specification of vestibular versus auditory organs.”

b) In Munnamalai et al., (2017), Wnt9a- over expression in the chicken ototcyst elicited some vestibular patches in the basilar papilla. The data presented by the authors here would suggest that somehow Wnt signaling was first suppressive to allow vestibular fates, before allowing being downregulated to allow a prosensory fate. But since it was already committed to behave as vestibular, it adopted a vestibular sensory fate. To align all these studies, this would suggest some kind of negative feed-back on the Wnt pathway itself is required to elicit vestibular patches within the basilar papilla as described.Following this thought, the authors show in Figure 4B, that T2-βcat-GOF overexpression shows no active Wnt signaling by E7. The authors suggest that those cells are “eliminated”. However, I would offer an alternative explanation that by E7, the Wnt pathway was actively down-regulated through negative feed-back. I would like to see one additional experiment where T2-βcat-GOF is over expressed and probed for Wnt-regulated, Wnt-inhibitors such as Axin2/ Nkd1 (PMCID-PMC4462952).

The overexpression of Wnt ligands may indeed produce a different response from that of βcat-GOF and we have made clear in the Discussion that the roles of Wnt signalling at a later stage of inner ear development (e.g patterning of the auditory organ of hair cell subtype specification, as highlighted in Munnamalai et al., 2017) change.

We would like to point out that in the experiments shown in Figure 4B, we are not monitoring the activity of the Wnt reporter, but a fluorescent protein constitutively expressed in the T2-βcat-GOF construct. Therefore, the expression of this fluorescent protein is not regulated by Wnt activity, or that of its inhibitors such as Axin2/Nkd1. It is possible that some form of negative feedback loop is triggered by the overexpression of βcat-GOF, but this was certainly not the case 24hours after electroporation since the Wnt reporter was clearly upregulated in transfected regions.

c) Are these ectopic patches (Figure 4) “vestibular” sensory patches? Use a vestibular marker such as gm2. This would help address if an intermediate level of Wnt is necessary for the formation of any prosensory epithelia? Or is it specific to cochlear sensory epithelia?

We did not test directly the vestibular “character” of these ectopic patches, but similarly to vestibular organs, the ectopic patches formed dorsally and were populated with numerous hair cells by E7 (hair cells form later in the auditory organ). Furthermore, some investigation into the nature of the ectopic sensory organs formed in chicken ears overexpressing βcat-LOF was already done by Stevens et al., (2003). In this study (performed with the same βcat-LOF coding sequence as in our experiments), the vestibular organs were fused with ectopic sensory patches harbouring vestibular hair cells.

[Editors' note: further revisions were suggested prior to acceptance, as described below.]

Essential revisions:1) Many questions were raised by the reviewers on the RNAseq results. After a thorough discussion, we recommend that the RNAseq data be removed from the current manuscript unless the authors could address the specific comments listed below.

We recognize the issues with variability of the two RNA-Seq experiments at the single gene level. We believe, however, that there is good reproducibility of the data at the “biological function” level, providing further evidence that Wnt signalling regulates a large set of neurosensory genes in the otocyst. The interpretation of these results or their significance is to some extent subjective and we think that exploring the role of a new candidate Wnt target gene is beyond the scope of the study, as is the validation of these RNA-Seq analyses through functional or expression studies. We have therefore decided to follow the reviewer’s recommendations and have omitted all RNA-Seq results from the present manuscript. As a consequence, Dr Vincent Plagnol who contributed exclusively to this aspect of the work is no longer a co-author in this revised version. We have also changed the order of the following results/figures:

– The effects of βcat-LOF on otic neurogenesis are now described earlier and in Figure 3—figure supplement 3.

– Figure 6 has been modified to include the effects of IWR-1 on *Sox2*/Jagged1 expression (Figure 6D-E’’), which were originally included in the figure describing the RNA-Seq screening experiments.

2) Provide a better quantification of the efficiency of downregulation of Sox2 by GOF Wnt activity.-The authors addressed the concern of lack of quantification by measuring fluorescence levels in two regions in two different otocysts, showing (in most cases in these chosen regions) an inverse correlation between transfected Wnt activity and Sox2 expression, as proposed. This correlation is interesting but does not address the concern that there were regions with GFP expression (reflecting high Wnt activity) and no apparent downregulation of Sox2. They acknowledged that there may be transgene expression differences. They changed the text to: The overexpression of βcat-GOF reduced, in a cell-autonomous manner, the levels of Jag1 and Sox2 expression in the majority of transfected prosensory cells but did not induce any change in the dorsal region of the otocyst (n=6)." However it is still unclear whether the Βcat-GOF decreased Sox2 in the majority of GFP-positive regions, since this quantification was not done.

We have now included further quantification of the effects of βcat-GOF on *Sox2* expression (see subsection “Quantification of *Sox2* expression” and new Figure 3—figure supplement 1, box plots in C and F). We used ImageJ to measure the average intensity levels of *Sox2* immunofluorescence in the individual nuclei of control (untransfected) and βcat-GOF transfected (GFP-expressing) cells located within the prosensory domains. The data were collected from two confocal stacks, representing a total of 509 untransfected cells versus 367 transfected cells. The results confirm a statistically significant down-regulation of *Sox2* expression in βcat-GOF transfected prosensory cells in both samples. We would like to point out that as shown by the box plots of the data, there is great variability in the endogenous levels of *Sox2* expression in control prosensory cells, as well as in the effects of βcat-GOF. It is therefore not possible to quantify the strength of the effects at the individual cell level (for example, “50% of transfected cells show a 50% reduction in *Sox2* expression”), since we do not know what were the original levels of *Sox2* expression in the cells that are analysed. We hope that this additional quantification addresses this reviewer’s comments.